# The aldehyde dehydrogenase 2 rs671 variant enhances amyloid β pathology

Xia Wang [1,3], Jiayu Wang [1,3], Yashuang Chen[1], Xiaojing Qian[2], Shiqi Luo[1], Xue Wang[2], Chao Ma [2] ✉ & Wei Ge [1] ✉

In the *ALDH2* rs671 variant, a guanine changes to an adenine, resulting in a dramatic decrease in the catalytic activity of the enzyme. Population-based data are contradictory about whether this variant increases the risk of Alzheimer's disease. In East Asian populations, the prevalence of the *ALDH2* rs671 variant is 30–50%, making the National Human Brain Bank for Development and Function (the largest brain bank in East Asia) an important resource to explore the link between the *ALDH2* rs671 polymorphism and Alzheimer's disease pathology. Here, using 469 postmortem brains, we find that while the *ALDH2* rs671 variant is associated with increased plaque deposits and a higher Aβ40/42 ratio, it is not an independent risk factor for Alzheimer's disease. Mechanistically, we show that lower ALDH2 activity leads to 4-HNE accumulation in the brain. The (*R*)−4-HNE enantiomer adducts to residue Lys53 of C99, favoring Aβ40 generation in the Golgi apparatus. Decreased ALDH2 activity also lowers inflammatory factor secretion, as well as amyloid β phagocytosis and spread in brains of patients with Alzheimer's disease. We thus define the relationship between the *ALDH2* rs671 polymorphism and amyloid β pathology, and find that *ALDH2* rs671 is a key regulator of Aβ40 or Aβ42 generation.

Aldehyde dehydrogenase 2 family member (ALDH2) is a key mitochondrial dehydrogenase that functions in the detoxification of acetaldehyde, and endogenous 4-hydroxy-2-nonenal (4-HNE) and malondialdehyde generated by polyunsaturated fatty acid peroxidation[1,2]. The *ALDH2* rs671 single-nucleotide polymorphism in exon 12 has received much research attention. This variant, G to A in the gene (NM_000690.4:c.1510G>A), causes a change from Glu to Lys (E to K) at position 504 of the ALDH2 protein (NP_000681.2:p.Glu504Lys), leading to a substantial decrease in dehydrogenase activity[1]. Please see the Glossary in Supplementary Notes for more information about the *ALDH2* rs671 polymorphism. The *ALDH2* A-allele is present in ~30–50% of the East Asian population (Chinese, Japanese, and Korean), compared with <5% of people of European ancestry[3]. Individuals with the GA and AA genotypes,

respectively, have an ALDH2 activity 10–45% and 1–5% of that in individuals with the GG genotype[3].

A large number of studies demonstrated an increased association between *ALDH2* rs671 polymorphism and many diseases[4,5], including an elevated risk of cancers[6] and cerebral vascular disease[7] after alcohol consumption, opposite effects in varied cardiovascular diseases[3], in *ALDH2* rs671 A-allele carriers. Recently, numerous studies focused on exploring whether *ALDH2* rs671 polymorphism elevates the risk of Alzheimer's disease (AD). However, population cohort studies have produced contradictory results[8–12]. Human Brain Bank allows postmortem analysis of brain pathology, including obtaining information on pathological grades [amyloid beta (Aβ) plaques, tau phosphorylation, neuritic plaques (Supplementary Methods)] and AD severity. International brain bank networks have been established, including

[1]Department of Immunology, Institute of Basic Medical Sciences Chinese Academy of Medical Sciences, School of Basic Medicine Peking Union Medical College, Beijing, China. [2]Department of Human Anatomy, Histology and Embryology, Neuroscience Center, National Human Brain Bank for Development and Function, Institute of Basic Medical Sciences Chinese Academy of Medical Sciences, School of Basic Medicine Peking Union Medical College, Beijing, China. [3]These authors contributed equally: Xia Wang, Jiayu Wang. ✉e-mail: machao@ibms.cams.cn; gewei@ibms.cams.cn

BrainNet Europe, the UK Brain Bank network, and others[13]. The National Human Brain Bank for Development and Function (hereinafter referred to as the Human Brain Bank, http://anatomy.sbm.pumc.edu.cn/brainbank) has developed rapidly in the past decade[14], with a collection of >750 whole brains by June 2023, becoming the largest brain bank in East Asia. Given the low incidence of the *ALDH2* rs671 variant in Western countries, and its high prevalence in East Asia, the Human Brain Bank is the ideal platform to recruit a large sample of brain tissues covering the *ALDH2* GG, GA, and AA genotypes to explore the relationships between *ALDH2* rs671 polymorphism and autopsy-confirmed AD pathology.

Here, we show that *ALDH2* rs671 variant affects Aβ pathology rather than susceptibility to AD and describes the underlying mechanisms. The data define the relationship between *ALDH2* rs671 polymorphism and AD.

## Results

### The *ALDH2* rs671 variant positively correlates with Aβ pathology, but not susceptibility to AD, based on samples from the Human Brain Bank

To determine the relationship between mitochondrial *ALDH2* rs671 polymorphism and AD pathology, a total of 469 participants from the Human Brain Bank were included for *ALDH2* rs671 sequencing, including 267 males (56.9%) and 202 females (43.1%). We excluded individuals with a history of long-term heavy alcohol consumption. Sanger sequencing results found 14 AA genotypes (3.0%), 123 GA genotypes (26.2%), and 332 GG genotypes (70.8%), with an overall A-allele frequency of 29.2% (Supplementary Fig. 1, Supplementary Data 1). A recent study from the China Kadoorie Biobank, including more than 500,000 participants from 10 study areas, reported that the overall rs671 A-allele frequency was 13–29% (varying by area)[7], consistent with our result. Spearman analysis demonstrated a lack of correlation of the rs671 genotype with sex ($P = 0.22$) (Supplementary Table 1). Considering the high correlation of age with AD pathology, ordinal logistic regression analysis with adjustment for age was conducted in the 469 individuals who were sampled for our study. The results showed that there was no significant effect of the GA/AA variants on the incidence of AD [estimated odds ratio (OR): 1.23, 95% confidence interval (CI): 0.84–1.80, $P = 0.29$]. However, the rs671 AA genotype was associated with a much higher Aβ plaque score compared with the GG genotype (OR: 3.35, 95% CI: 1.25–8.98, $P = 0.02$) (Fig. 1a). The A score reflects the brain regions extent of Aβ plaque appearance in the brain. This result suggested a more extensive spread of plaques in AA genotype brains. Further analysis stratified by sex, we observed that the rs671 AA genotype exacerbated Aβ pathology in males ($n = 267$), the effect was not statistically significant in females ($n = 202$) (Supplementary Table 2). No higher risk was found among GA or AA genotypes for tau hyperphosphorylation, neuritic plaque accumulation, average Everyday Cognition (ECog) score (Fig. 1a), or other co-neuropathologies including Lewy bodies, Braak staging of Parkinson's disease, TDP-43 pathology, Primary age-related tauopathy, and Cerebral amyloid angiopathy (Supplementary Table 3) (Neuropathological evaluation—Supplementary Methods).

To determine if the *ALDH2* rs671 A-allele is positively associated with an increased number of Aβ plaques, we immunostained amyloid plaques in eight regions from 44 postmortem brains with AD (18 with genotype GG, 18 GA, and 8 AA) (Supplementary Table 4, Supplementary Data 1). The eight brain regions included the inferior parietal lobule (IPL), middle frontal gyrus (MFG), superior temporal gyrus (STG), hippocampus (Hipp), basal ganglia (BG), visual cortex (VC), midbrain (Mid), and cerebellum (Cblm); these regions were used to determine A scores (which range from 0 to 3) according to National Institute on Aging/Alzheimer Association guidelines[15] (Fig. 1b). Notably, an increased Aβ plaque area was found in rs671 GA and AA groups in various regions of the brain compared with the GG group (Fig. 1c,

Supplementary Fig. 2a). This observation was consistent for both males ($n = 21$) and females ($n = 23$) (Supplementary Table 5).

Aβ peptides of varied lengths, from 37 to 49 amino acids, are generated by sequential proteolytic cleavage of amyloid beta precursor protein (APP) by β-secretase and the γ-secretase complex (see Glossary in Supplementary Notes). The longer peptide Aβ42 (having 42 amino acids) is thought to be more toxic and prone to aggregation than the shorter Aβ40[16]. A changed ratio of Aβ42 to Aβ40 is a strong indicator of AD[17,18]. Therefore, we assessed Aβ40 and Aβ42 levels in MFG, STG, and Hipp from 44 postmortem brains by enzyme-linked immunosorbent assay (ELISA) (Supplementary Table 4, Supplementary Data 1). The data showed an elevation of Aβ40 peptides ($P < 0.05$) and a decrease of Aβ42 peptides ($P < 0.05$) in rs671 GA/AA carriers, ultimately leading to higher Aβ40/42 ratios ($P < 0.05$) in rs671 GA/AA carriers than in GG carriers (Fig. 1d–f), both in males ($n = 21$) and females ($n = 23$) (Supplementary Table 6). However, the rs671 variants did not alter the RNA or protein expression levels of APP or presenilin 1 (PS1, the main catalytic subunit of γ-secretase) (Supplementary Fig. 2b, γ-secretase—see Glossary in Supplementary Notes).

Notably, immunoblotting with anti-ALDH2 revealed lower ALDH2 protein levels in postmortem specimens with the rs671 GA or AA genotype (Fig. 1g, Supplementary Data 1). Proteomic analysis of 27 postmortem human hippocampal extracts, including 15 GG carriers, 6 GA, and 3 AA, confirmed the lower ALDH2 protein level (Supplementary Fig. 2c, Supplementary Data 1). Representative MS/MS profiling indicated significantly lower ALDH2 peptide levels in GA/AA individuals (Supplementary Fig. 2d). However, transcriptome sequencing conducted on 50 postmortem human hippocampal tissues[19], comprising 35 GG carriers, 13 GA, and 2 AA (Supplementary Table 7, Supplementary Data 1), detected comparable *ALDH2* mRNA levels between groups (Supplementary Fig. 2e). These results are in agreement with a previous report, which showed that the replacement of Glu487 by Lys487 caused dramatic lower ALDH2 protein levels in human and mouse liver due to the decreased stability of the protein[20].

We also sequenced alcohol dehydrogenase 1B (class I), beta polypeptide (*ADH1B*) rs1229984 in 190 enrolled postmortem brains (Supplementary Data 1). ADH1B is the first enzyme in alcohol metabolism. The rs1229984 T variation enhances the activity of ADH1B and accelerates the conversion of alcohol to acetaldehyde. The results showed 75 rs1229984 TT carriers (39.5%) and 88 CT (46.3%), with an overall T-allele frequency of 85.8% (Supplementary Fig. 3), similar to the previous finding of 90% among Taiwanese[21]. Ordinal logistic regression analysis demonstrated that the *ADH1B* rs1229984 variant was not associated with susceptibility to AD pathology (Supplementary Fig. 4).

Thus, collectively, these results demonstrate that ALDH2 activity deficiency, but not enhanced ADH1B activity, augments Aβ pathology in human brains, evidenced by increased amyloid plaque load and a higher Aβ40/42 ratio, indicating the *ALDH2* rs671 G>A variant is not an independent risk factor for AD.

### The increased Aβ40/42 ratio is dependent on ALDH2 activity

To address if the alteration of Aβ pathology is dependent upon ALDH2 activity, we assessed the Aβ40 and Aβ42 peptide levels in the brains of *Aldh2*-knockout (*Aldh2*[-/-]) mice and Aldh2-activity-inhibited APP/PS1 mice (APP/PS1 mice, see the Glossary in Supplementary Notes for more details about the mice). The results showed that *Aldh2* knockout (Fig. 2a, b) significantly increased the Aβ40/42 ratio in the COR (1.48-fold, $P < 0.05$) (Fig. 2c) and Hipp (1.83-fold) (Supplementary Fig. 5a), with higher Aβ40 levels and no obvious fluctuation of Aβ42 levels, in brains of *Aldh2*[-/-] mice compared with age-matched wild-type (WT) C57BL/6 mice. Altered Aβ pathology was also found in APP/PS1 mice after treatment with the ALDH2 inhibitor daidzin (150 mg/kg/day for 2 months) (Fig. 2d), which suppressed Aldh2 enzyme activity[22]. The results demonstrate that suppressed Aldh2 activity markedly elevated

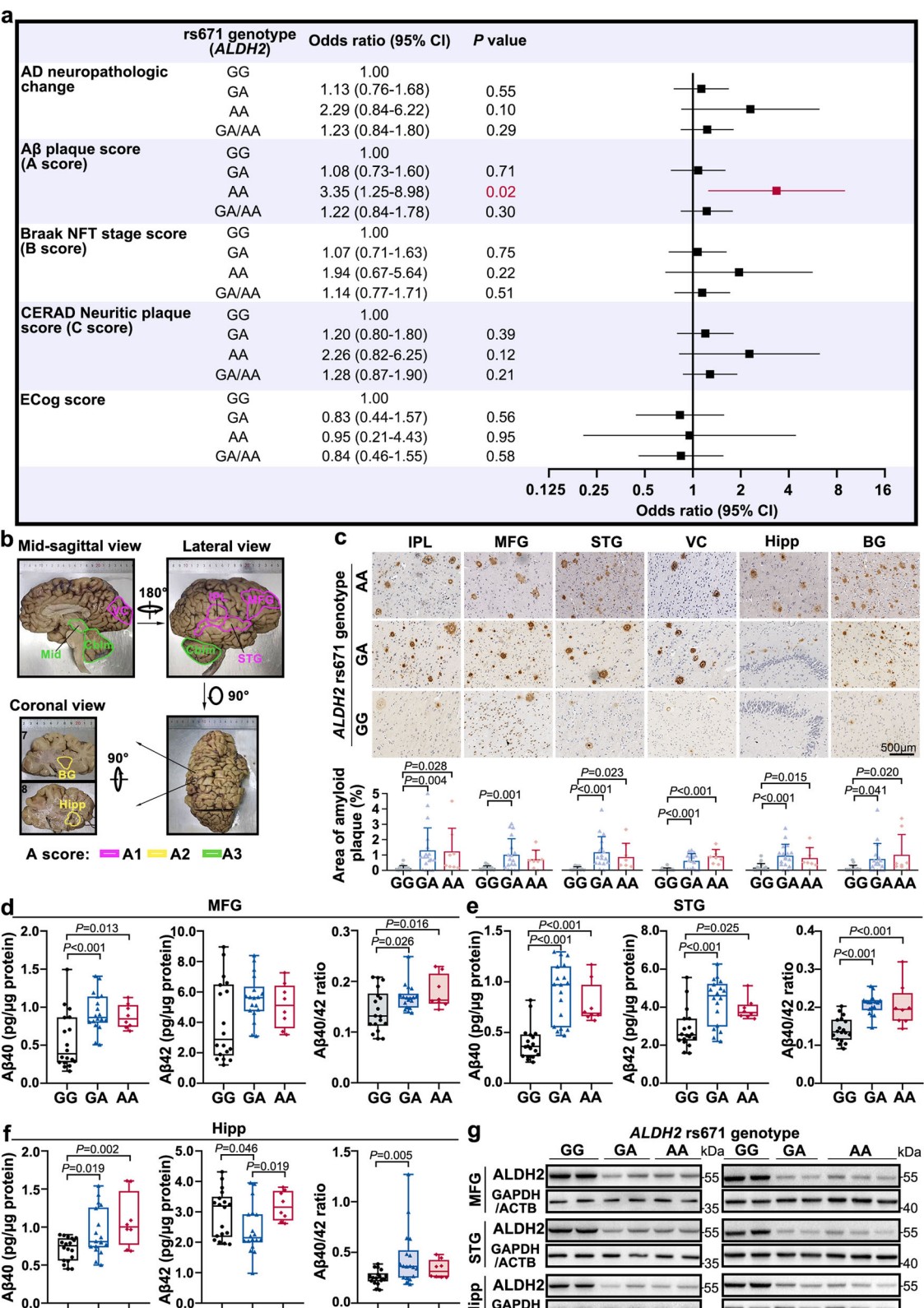

Aβ plaque load in the COR and Hipp (Fig. 2e) and enhanced the Aβ40/42 ratio (1.78-fold, $P < 0.05$), with an increase of Aβ40 and a decrease of Aβ42 (Fig. 2f).

Next, we employed RNA silencing (siALDH2) or daidzin treatment to decrease the ALDH2 activity in the neuronal cell models SH-SY5Y and N2a-APPswe [Neuro-2a cells stably expressing APP with the so-called "Swedish mutation" (K670N/M671L), develops autosomal

dominant familial AD with an onset age of 45–60 ages[23]], and HEK293T cells. Quantitative real-time PCR (RT-qPCR) and western blot (WB) confirmed the downregulation of ALDH2 at the mRNA and protein levels after ALDH2 silencing (Fig. 2i–j, Supplementary Fig. 5b, c, g, h, k, l). Lower ALDH2 enzymatic activity was validated after RNA silencing or daidzin treatment (Fig. 2g, Supplementary Fig. 5i, m). ELISA or WB showed that lower ALDH2 activity increased the Aβ40/42 ratio in all

**Fig. 1 | Mitochondrial aldehyde dehydrogenase 2 (*ALDH2*) rs671 A-variant positively correlates with exacerbated amyloid plaque pathology in human brains. a** Association of risk factors with *ALDH2* rs671 polymorphism and Alzheimer's disease (AD)-related neuropathologic changes after adjustment for age in Chinese populations. Odds ratios and *P* values were calculated by ordinal logistic regression with adjustment of age using SPSS software. *n* = 469. **b** Representative image of the anatomy of the right human brain with formalin fixation. Eight brain regions were immunostained for amyloid-β (Aβ) plaque deposition to assess Aβ pathology according to National Institute on Aging/Alzheimer Association guidelines for the neuropathologic assessment of AD. **c** Representative images of Aβ deposits labeled with 6E10 antibody (anti-β-amyloid 1–16 antibody) in six subregions from postmortem brains with pathological AD with different rs671 genotypes. rs671 GG genotype (*n* = 18, 82.89 ± 7.76 yr), GA (*n* = 18, 84.78 ± 4.40 yr), AA (*n* = 8, 84.62 ± 9.76 yr). Percentage of amyloid plaque area was determined. Scale bar, 500 μm. **d–f** Quantification of Aβ42 and Aβ40 peptides in frozen (**d**) MFG homogenates, (**e**) STG homogenates, and (**f**) Hipp homogenates from the above-mentioned 44 postmortem brains with pathological AD, determined by enzyme-linked immunosorbent assay (ELISA). rs671 GG genotype (*n* = 18, 82.89 ± 7.76 yr), GA (*n* = 18, 84.78 ± 4.40 yr), AA (*n* = 8, 84.62 ± 9.76 y). One datapoint represents one sample per genotype. All box plots include the median line, the box indicates the interquartile range, and whiskers indicate minima and maxima. **g** Western blot (WB) with anti-ALDH2 in frozen MFG, STG, and Hipp subregions from postmortem brains with pathological AD with rs671 GG (81.50 ± 1.91 yr), GA (84.25 ± 7.85 yr), and AA (90.40 ± 2.79 yr) genotypes. *n* = 4 or 5. IPL inferior parietal lobule, MFG middle frontal gyrus, STG superior temporal gyrus, Hipp hippocampus, BG basal ganglia, VC visual cortex. Data are presented as mean values ± SD. Statistical analysis was performed using one-way analysis of variance (ANOVA) with least significant difference (LSD) post-hoc test for multiple groups. Source data are provided as a Source Data file.

three cell lines by either increasing Aβ40 or decreasing Aβ42 production (Fig. 2h, k, Supplementary Fig. 5j, n). In SH-SY5Y cells, daidzin treatment increased the Aβ40/42 ratio by decreasing Aβ42 production without affecting Aβ40 production (Fig. 2h). In HEK293T cells, combined *ALDH2* knockdown and daidzin treatment further decreased Aβ42 production and increased the Aβ40/42 ratio (Fig. 2k). These results indicate that ALDH2 modulates the Aβ40/42 ratio in an enzyme activity-dependent manner.

Taken together, increased Aβ deposits and Aβ40/42 ratio were evident in *Aldh2*$^{-/-}$ mice brains, daidzin-treated APP/PS1 mice brains, and ALDH2-activity-deficient cells, indicating that loss of ALDH2 function is sufficient to exacerbate Aβ pathology.

## Lower ALDH2 activity elevates the Aβ40/42 ratio mainly via (*R*)−4-HNE mediation

To elucidate the molecular mechanisms underlying the neuronal effects of Aldh2 deficiency, we performed high-throughput proteomics to identify changes in protein expression caused by daidzin treatment in N2a-APPswe cells (Supplementary Fig. 6a). A total of 2900 proteins were detected with stringent criteria for high protein confidence (unique peptides >1 and false discovery rate ≤ 0.01) (Supplementary Fig. 6b). Compared with untreated N2a-APPswe cells, daidzin treatment caused differential expression of 100 proteins, with 36 downregulated and 64 upregulated (Supplementary Fig. 6c, Supplementary Data 2). KEGG pathway analysis of the differentially expressed proteins (DEPs) revealed dysregulated metabolic pathways and fatty acid metabolism (Fig. 3a). Therefore, the proteomic data suggest that suppression of Aldh2 enzyme activity could impair fatty acid metabolism in neuronal cells.

ALDH2 plays a critical role in detoxifying endogenous aldehydes such as 4-HNE and malondialdehyde produced from lipid peroxidation triggered by oxidative stress[1,24,25]. As the main enzyme for 4-HNE metabolism, deficiency of ALDH2 activity impedes 4-HNE clearance, leading to higher 4-HNE levels in diverse disease states[24]. We hypothesized that the increased Aβ40/42 ratio caused by lower ALDH2 activity may be mediated by accumulated 4-HNE. We thus examined 4-HNE levels with 4-HNE polyclonal antibody and found increased accumulation of 4-HNE adducts in the Hipp of postmortem brains with the rs671 GA/AA genotypes (Fig. 3b), in the neocortex of *Aldh2*$^{-/-}$ mice (Fig. 3c), and in SH-SY5Y cells with ALDH2 activity deficiency (Fig. 3d, Supplementary Fig. 7a).

4-HNE is a racemate, with chirality at carbon 4. Different cellular responses have been reported on treatment with (*R*)−4-HNE, (*S*)−4-HNE, and (±)−4-HNE (i.e., the racemic mixture). For example, (*S*)−4-HNE displayed more toxicity than (*R*)−4-HNE, with greater inhibition of cell proliferation and more apoptosis in rat Clone 9 cells[26,27]. Here, this study aimed to determine if 4-HNE regulates the Aβ40/42 ratio and which configuration of 4-HNE is more important.

(*R*)−4-HNE was obtained by two methods in this study: in the first, standard (*R*)−4-HNE was obtained from its stable precursor (*R*)−4-hydroxynonanal dimethylacetal (Avanti, 870608) after deprotection by 1 mM HCl-catalyzed transacetalization (Supplementary Fig. 7b); in the second, (*R*)−4-HNE was purified (>95%) (Fig. 3e) from racemic (±)−4-HNE by chiral separation using high-performance liquid chromatography (HPLC). The second method also produced (*S*)−4-HNE (>95% purity), which eluted earlier than (*R*)−4-HNE (Supplementary Fig. 7c).

Cell counting kit-8 (CCK8) assay of SH-SY5Y cells showed that doses of (±)−4-HNE ≥ 5 μM greatly inhibited cell proliferation after prolonged treatment (>12 h for 10 μM, and >24 h for 5 μM); lower doses (<5 μM) had no obvious effect even after treatment for 48 h (Supplementary Fig. 7d). The conversion of LC3-I to LC3-II is widely used to assess autophagy of cells[28]. Therefore, we detected LC3-II levels in SH-SY5Y cells to assess autophagy induced by 4-HNE. Propidium iodide (PI, an end-stage apoptosis marker) and fluorescein isothiocyanate (FITC)-conjugated Annexin V (Annexin V-FITC) were used as labels to monitor the progression of apoptosis. Treatment of SH-SY5Y cells with 2 μM 4-HNE did not trigger an apoptotic or autophagic response, even after 72 h (Fig. 3f, g, Supplementary Fig. 7e). However, 10 μM 4-HNE for 4 h induced obvious autophagy in SH-SY5Y, as evidenced by increased levels of LC3A-II (Fig. 3f); similar effects were observed for (±)−4-HNE, (*R*)−4-HNE, and (*S*)−4-HNE. In the following experiments, two conditions were applied to SH-SY5Y cells: acute treatment with 10 μM 4-HNE for 4 h or long-term treatment with 2 μM 4-HNE for 24 h.

Next, we investigated the effects of 4-HNE on Aβ40 and Aβ42 production in neuron cells by ELISA. Acute treatment to different concentrations of (±)−4-HNE (0, 1, 2.5, 5, 10 μM) or (*S*)−4-HNE (5, 10 μM) for 4 h did not alter the Aβ40/42 ratio in SH-SY5Y cells (Fig. 3h, Supplementary Fig. 7f). However, 10 μM (*R*)−4-HNE for 4 h significantly increased the Aβ40/42 ratio (*P* < 0.01) in SH-SY5Y, with marked elevation of Aβ40 and reduction of Aβ42 levels, compared with phosphate-buffered saline (PBS)-treated control cells (Fig. 3h). This effect of (*R*)−4-HNE on Aβ40/42 ratio elevation in SH-SY5Y cells was confirmed using standard (*R*)−4-HNE derived from (*R*)−4-hydroxynonanal dimethylacetal (Supplementary Fig. 7g). Similarly, an increased Aβ40/42 ratio was found in SH-SY5Y cells treated by long-term (24 h) of 2 μM (*R*)−4-HNE. Compared with controls, all three forms of 4-HNE at 2 μM inhibited both Aβ40 and Aβ42 levels. However, only (*R*)−4-HNE treatment resulted in enhanced inhibition of Aβ42 production and consequent elevation of the Aβ40/42 ratio (Fig. 3i). Therefore, deficient ALDH2 activity induces a higher Aβ40/42 ratio in neuronal cells, mainly mediated by (*R*)−4-HNE.

## (*R*)−4-HNE enhances the Aβ40/42 ratio by covalent adduction on C99

Aβ peptides are generated by γ-secretase cleavage of substrate C99, the *C*-terminal 99-residue fragment of APP (Aβ peptides production, see Glossary in Supplementary Notes). To examine the

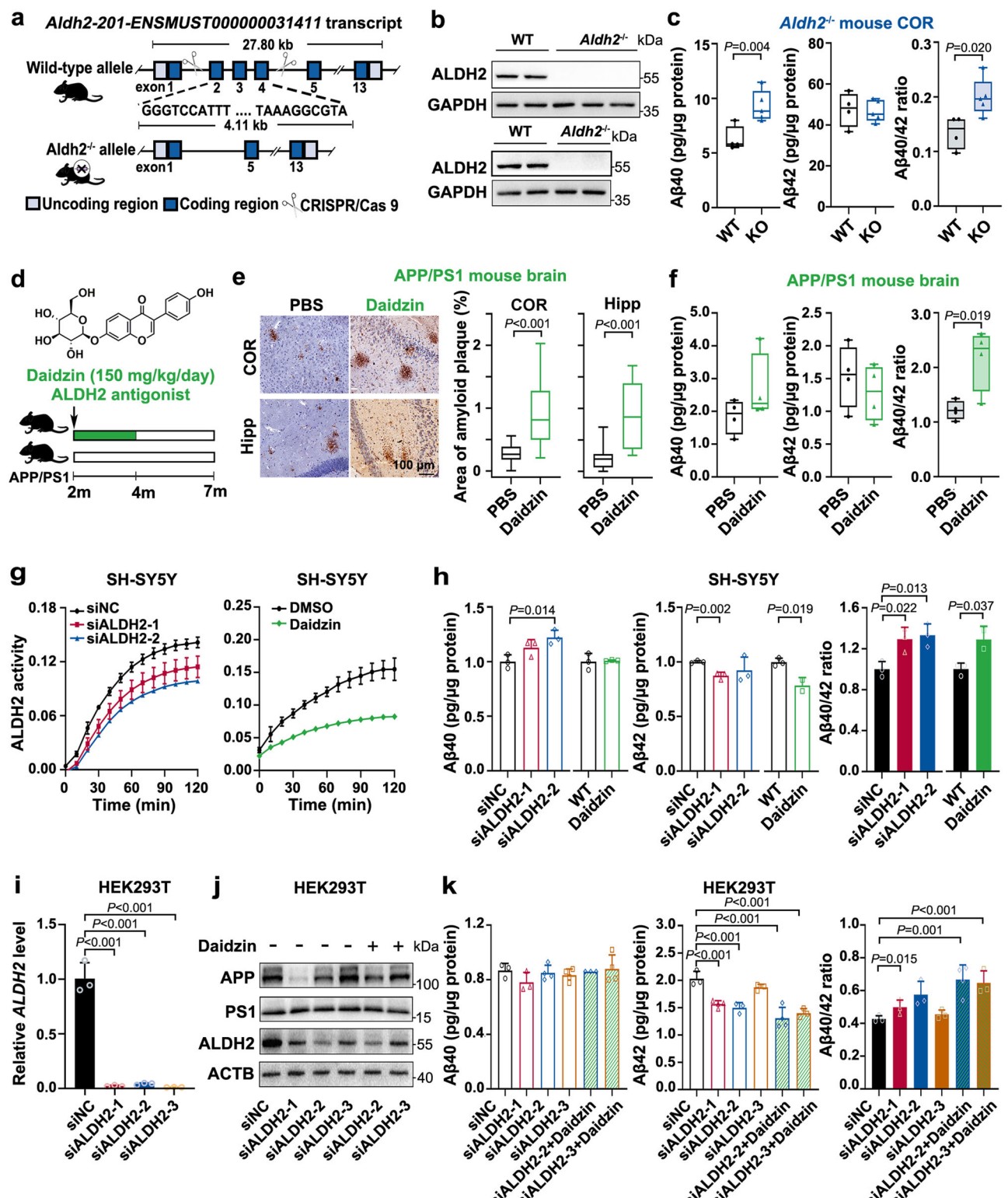

effect of $(R)$−4-HNE on Aβ generation-related biochemical changes, we prepared 4-HNE-modified substrate and 4-HNE-modified γ-secretase complex, followed by γ-secretase cleavage assay and detection of Aβ40 and Aβ42. Briefly, substrates (natural or 4-HNE-modified) were obtained from N2a-APPswe cells pretreated without or with 10 μM $(R)$−4-HNE for 4 h. The γ-secretase complex was extracted from HEK293T cells pretreated without or with 10 μM $(R)$−4-HNE for 4 h, as previously described[29]. The result indicates that more 4-HNE adducts were detected in the $(R)$−4-HNE-treated

groups than in their PBS-treated (WT) counterparts (Supplementary Fig. 8a).

Following the γ-secretase cleavage assay conducted at 37 °C for 12 h, Aβ40 and Aβ42 levels were determined by ELISA. Compared with the natural γ-secretase, the 4-HNE-adducted γ-secretase decreased both Aβ42 and Aβ40 production (third column vs. first column: 0.71-fold for Aβ42 and 0.73-fold for Aβ40; fourth column vs. second column: 0.80-fold for Aβ42 and 0.75-fold for Aβ40) (Fig. 4a), but did not alter the Aβ40/42 ratio. Compared with natural substrate, the 4-HNE-

**Fig. 2 | Reduced ALDH2 activity significantly increased the Aβ40/42 ratio.**
**a** Schematic image of the endogenous loci to generate *Aldh2*-knockout mice in a C57BL/6 background. (Image was created using Photoshop CS6, with elements from ChemBioDraw Ultra software 14.0.) **b** WB confirmed the Aldh2 expression in *Aldh2*$^{-/-}$ mouse brains. All mice with *Aldh2*$^{-/-}$ showed similar results. **c** Levels of Aβ42 and Aβ40 peptides in cortex (COR) homogenates from 3-month-old *Aldh2*$^{-/-}$ mice (*n* = 5 mice) and age-matched wild-type mice (*n* = 4 mice) by ELISA. **d** Scheme illustrating intragastric administration of daidzin (150 mg/kg/day) in 2-month-old APP/PS1 mice for 2 months. Aβ pathology was assessed in 7-month-old mice. (Image was created using Photoshop CS6 and ChemBioDraw Ultra software 14.0.) **e** Immunohistochemical staining of Aβ plaque deposits in COR and Hipp and **f** quantification of Aβ40 and Aβ42 levels in COR from APP/PS1 transgenic mice with phosphate-buffered saline (PBS, *n* = 4 mice) or daidzin administration (*n* = 4 mice). **g** and **h** ALDH2 knockdown by siRNA or daidzin (30–100 μM) treatment for 48 h in

SH-SY5Y cells. **g** Enzymatic activity of ALDH2 in cell lysates, measured over 120 min by using a mitochondrial aldehyde dehydrogenase (ALDH2) Activity Assay Kit (Abcam, ab115348). *n* = 3 biologically independent samples. **h** Levels of intracellular Aβ42 and Aβ40 peptides. *n* = 3 biologically independent samples. **i**–**k** ALDH2 functional deficiency caused by ALDH2 silencing or combined with daidzin co-treatment for 48 h in HEK293T cells. **i** RT-qPCR of relative *ALDH2* mRNA levels (*n* = 3 biologically independent experiments) and **j** WB detection of ALDH2, APP, and PS1 expression levels. **k** Levels of intracellular Aβ42 and Aβ40 peptides. *n* = 3 or 4 biologically independent samples. siNC negative control small interfering RNA. Data are presented as mean values ± SD. All box plots include the median line, the box indicates the interquartile range, and whiskers indicate minima and maxima. Statistical analysis was performed using a two-tailed Student's *t*-test for two groups (**c**, **e**, **f**, **h**, **i**, **k**) and one-way ANOVA with LSD post-hoc test for multiple groups (**h**). Source data are provided as a Source Data file.

adducted substrate decreased Aβ42 production (second column *vs.* first column: 0.72-fold; fourth column vs. third column: 0.81-fold), but not Aβ40 levels, and thus increased the Aβ40/42 ratio (second column vs. first column: 1.47-fold; fourth column vs. third column: 1.42-fold) (Fig. 4a). These results demonstrate that (*R*)−4-HNE increased the Aβ40/42 ratio by adducting substrate C99. We then verified the 4-HNE-modification of APP in substrate extracts using immunoprecipitation (Fig. 4b).

To further test the impact of 4-HNE modification of substrate on Aβ40 and Aβ42 generation, we purified recombinant human C99 with a 6×His-tag at the *C*-terminus and treated it with (*R*)−4-HNE. The result indicates an increase of C99 dimer and trimer formation after 4-HNE-adduction, indicating that 4-HNE induced enhanced oligomerization (Supplementary Fig. 8b). γ-Secretase cleavage assays revealed that 4-HNE-modification of C99 shifted the Aβ40/42 ratio toward Aβ40 generation (Fig. 4c). This finding corroborates the data in Fig. 4a.

Protein carbonylation, occurring as a consequence of the generation of adducts with Lys, Cys, and His residues, has been recognized as one of the preeminent toxic actions of 4-HNE[30]. There are 13 potential 4-HNE reaction sites (Lys, His) in C99, including Lys53–Lys54–Lys55, which are near residues Thr48 and Leu49, the primary cleavage site of γ-secretase (Fig. 4d). According to the cryo-electron microscopy (cryo-EM) structure of human γ-secretase in complex with a C99 fragment (PDB: 6IYC)[31], free C99 (PDB: 2LLM)[32] undergoes remodeling on binding to γ-secretase, with residues Met51–Lys53 forming a β-strand. The formation of this β-strand is required for the primary endopeptidase cleavage of C99 (which generates Aβ49 or Aβ48) via the formation of a hybrid β-sheet with a β-strand from PS1[29,33] (Fig. 4e). Therefore, Lys53 is a critical residue in regulating γ-secretase cleavage. Two types of chemical reactions can occur between 4-HNE and a Lys residue: either the 2′ C=C double bond of 4-HNE reacts with the Lys residue via Michael addition, or the 4′ aldehyde group reacts with the Lys residue via Schiff-base formation[30] (Fig. 4f). From tandem mass tag (TMT)-labeled proteomic data for plaques from postmortem brains with pathological AD and healthy control individuals, we detected 4-HNE adduction on C99 residue Lys53 via Schiff-base formation (PXD005824) (Fig. 4g).

### (*R*)−4-HNE adduction on Lys53 via Schiff-base formation causes positional deviation of C99, leading to preferential generation of Aβ40

To further explore the regulation of Aβ40 and Aβ42 generation via the reaction at site Lys53, we synthesized a short peptide with *N*-terminal acetylation, Ac-Leu–Lys–Lys–Lys–Gln (residues 52–56 of C99). The chemical structure and MS spectra of Ac-Leu–Lys–Lys–Lys–Gln are shown in Supplementary Fig. 9a. After reaction for 1 h at room temperature on the addition of a 50-fold (molar ratio) of (*R*)−4-HNE, matrix-assisted laser desorption/ionization-time of flight mass spectrometry (MALDI-TOF MS) detected a product with a molecular weight increase of 177.14 Da, namely 138.04 (Schiff-base formation) + 39.10 (a

potassium ion adduct) (Fig. 5a). The MS/MS analysis revealed that the primary site of adduction reaction was the first Lys residue (Lys53 of C99) within the peptide, accounting for the vast majority of the adduction reaction (Fig. 5a). A small proportion of adduct via Michael addition on −NH$_2$ of the second Lys (Lys54 of C99) was also observed (molecular weight increase of 156.11 Da) (Supplementary Fig. 9).

Further, we expressed a recombinant form of C99 with mutation Lys53Arg, namely C99$^{Lys53Arg}$, which abrogated 4-HNE modification at residue 53. After the reaction of C99$^{Lys53Arg}$ with (*R*)−4-HNE, WB detected modified peptides mainly as dimers (Fig. 5b). In in vitro γ-secretase cleavage assay, (*R*)−4-HNE modification of C99 significantly increased Aβ40 generation (1.5-fold) and slightly lowered Aβ42 generation compared with natural C99. Compared with untreated C99$^{Lys53Arg}$, (*R*)−4-HNE modification of C99$^{Lys53Arg}$ significantly lowered γ-secretase cleavage efficiency, with both decreased Aβ40 (0.46-fold) and Aβ42 (0.34-fold) generation. The Lys53Arg mutation attenuated the (*R*)−4-HNE-induced Aβ40/42 ratio increase from 1.73-fold (*P* < 0.05) to 1.32-fold (not statistically significant) (Fig. 5c).

ZDOCK protein modeling software[34] was used to predict binding interactions between a 4-HNE modified [Val46–Lys53(HNE)–Lys55] or natural [Val46–Lys55] truncates of C99 with γ-secretase. The predicted structures indicated a mild positional deviation of 4-HNE-modified Lys53(HNE) compared with the natural Lys53, towards the cave between transmembrane (TM) 8 and TM1, possibly due to 4-HNE adduction-induced steric hindrance (Fig. 5d). The 4-HNE modification also caused the distance change between Leu49 of C99 and the carboxylate side chain of Asp385, the catalytic residue of PS1. The distance between the carboxylate side chain of Asp385 and the C=O group of Leu49 decreased from 5.0 Å in natural peptide to 3.9 Å in 4-HNE-modified peptide, while the distance between the carboxylate side chain of Asp385 and the NH group of Leu49 from 6.2 Å in natural peptide to 5.5 Å in the 4-HNE-modified peptide (Fig. 5d, Supplementary Fig. 10). The closer distance of 3.9 Å indicated an increased propensity of the amide bond between Leu49 and Val50 to be hydrolyzed by Asp385, thus preferring generation of Aβ49 and Aβ40.

### (*R*)−4-HNE-adducted APP aggravates Aβ40 generation by retrograding to the Golgi apparatus

It has been reported that Aβ production is found in multiple intracellular sites, such as the endoplasmic reticulum (ER), Golgi apparatus, mitochondria, endosomes, lysosomes, multivesicular bodies, and the cytosol. Different organelles show different preferences for Aβ40 and Aβ42 generation. To explore whether (*R*)−4-HNE modification altered the intracellular location of APP processing, we performed immunofluorescence microscopy to visualize APP colocalization with different organelle markers (RCAS1 for the Golgi apparatus, LAMP2b for lysosomes, CANX for ER, and Rab5 for early endosomes) in HEK293T cells after (*R*)−4-HNE treatment. Double staining revealed coinciding fluorescence signals for C1/6.1 (an antibody for the APP *C*-terminal fragment) and RCAS1 in (*R*)−4-HNE-treated HEK293T cells, indicating

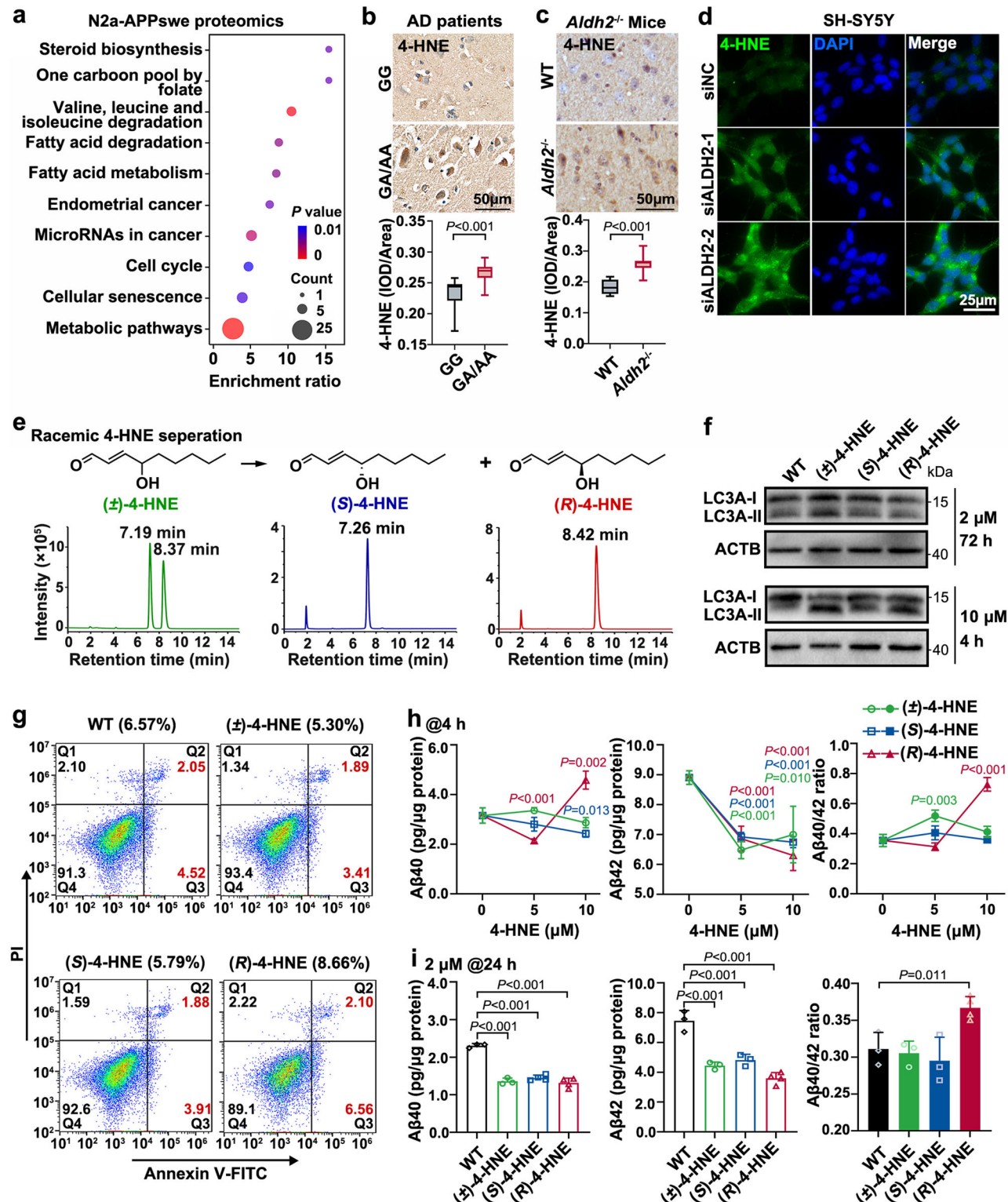

promoted APP localization in the Golgi apparatus (Fig. 6a, Supplementary Fig. 11a). APP+ puncta were rarely found to be localized in early endosomes, lysosomes, or the ER (Supplementary Fig. 11b). To explore whether 4-HNE modification altered Aβ40 and Aβ42 generation in the Golgi apparatus, we then isolated Golgi apparatus using a commercial kit (Invent, GO-037) from HEK293T cells (Supplementary Fig. 11c). Obvious enhanced expression of APP and γ-secretase components (PS1, APH1, PEN2) was observed in Golgi apparatus homogenates from (R)−4-HNE-pretreated HEK293T cells compared with their untreated counterparts (Fig. 6b). ELISA data showed that 4-HNE modification

significantly increased Aβ40 generation in the Golgi homogenates, but there was almost no change in Aβ42 generation, leading to a much higher Aβ40/42 ratio (Fig. 6c). This may be caused by the increase of 4-HNE-modified APP in the Golgi apparatus.

The trans-Golgi network (TGN) is a favorable subcellular site for Aβ40 generation. Endocytosed APP recycling from early endosomes to the TGN mediated by the retromer complex is essential for efficient Aβ40 production[35]. Several studies support the APP sorting mechanism in which SORL1 acts as a bridging receptor[36,37]. SORL1 directly interacts with APP through the CR-cluster and binds to the

**Fig. 3 | Reduced ALDH2 activity induced 4-hydroxy-2-nonenal (4-HNE) accumulation, and only the chiral (R)−4-HNE led to Aβ40/42 ratio increase.**
**a** Proteomic profiling was carried out using N2a-APPswe cells pretreated with PBS or 60 μM daidzin for 48 h. KEGG pathways analysis of the differentially expressed proteins (DEPs). **b** Representative images of 4-HNE immunohistochemical staining of paraffin sections of the human hippocampus with pathological AD. n = 3 per group. rs671 GG (86.33 ± 7.02 yr), GA/AA (82.33 ± 1.15 yr). Scale bar, 50 μm.
**c** Representative images of 4-HNE immunostaining of frozen brain sections of 3-month-old male *Aldh2*⁻/⁻ mice (n = 5) and wild-type mice (n = 4). Scale bar, 50 μm.
**d** Representative images of immunocytochemical staining with 4-HNE antibody (green) in SH-SY5Y cells after ALDH2 silencing. Three independent biological replicates were performed. Scale bar, 25 μm. **e** Separation of (R)−4-HNE and (S)−4-HNE from (±)−4-HNE by high-performance liquid chromatography (Shimadzu) with a chiral column (4.6 × 150 mm, CHIRALPAK AS-H). **f** and **g** SH-SY5Y cells were pretreated with PBS, or (±)−4-HNE, (R)−4-HNE, or (S)−4-HNE at 2 μM for 72 h or 10 μM for 4 h. **f** Autophagy was assessed by WB for LC3A-I/LC3A-II, the typical marker of autophagy. **g** Apoptosis was assessed by flow cytometry. The flow-cytometry gating strategies were shown in Supplementary Fig. 7e. **h** and **i** Levels of intracellular Aβ40 and Aβ42 were determined by quantitative ELISA. **h** SH-SY5Y cells were treated with 0, 5, or 10 μM (±)−4-HNE, (R)−4-HNE, or (S)−4-HNE for 4 h. **i** SH-SY5Y cells were treated with 2 μM (±)−4-HNE, (R)−4-HNE, or (S)−4-HNE for 24 h. n = 3 biologically independent samples. Data are presented as mean values ± SD. All box plots include the median line, the box indicates the interquartile range, whiskers indicate minima and maxima. Statistical analysis was performed using two-tailed Student's *t*-test for two groups (**b** and **c**) and one-way ANOVA with LSD post-hoc test for multiple groups (**h** and **i**). Source data are provided as a Source Data file.

retromer subunit VPS26 via a cytoplasmic FANSHY sequence. The VPS26–VPS35–VPS29 trimer is the core of the retromer complex, where VPS35 forms a platform for VPS26 and VPS29 to assemble[38] (Fig. 6d). We determined that 4-HNE treatment did not alter the expression level of SORL1 in HEK293T cells. However, immunoprecipitation suggested more 4-HNE-modified APP interacting with SORL1 than the unmodified APP (Fig. 6e). That is, SORL1 can traffic more 4-HNE-modified APP to the TGN. Mehmedbasic et al. found that the SORL1 CR(5-8) exhibits a higher affinity for binding to APP, compared to CR(1-4) or CR(1-8)[39], although the precise underlying mechanisms remain elusive. In this study, we predicted protein-protein docking between APP770 and YWTDs–CR–FN domains of SORL1 by Rosseta-Dock. The resultant docking results revealed a direct interaction between the *N*-terminal Met1–Val20 of APP (green) and SORL1 CR(5−8) (purple). Notably, the presence of the CR(1−4) domain introduces steric hindrance, thereby impeding access of APP N-tail to the binding site within the SORL1 CR(5−8) domain (Fig. 6f). Collectively, this structural elucidation provides a plausible rationale for the binding interaction between SORL1 and APP. In addition, we found ALDH2 knockdown did not alter VPS35 expression at either the mRNA or protein level (Fig. 6g). However, depletion of VPS35 in HEK293T cells (Fig. 6h) attenuated the 4-HNE-induced redistribution of APP to the Golgi apparatus (Fig. 6i, j, Supplementary Fig. 11f), suggesting that VPS35 mediates recycling of 4-HNE-modified-APP to the Golgi apparatus from early endosomes, perhaps because of higher transport efficiency of 4-HNE-modified APP than of natural APP. VPS35 knockdown interrupted 4-HNE-induced APP processing in the Golgi apparatus of HEK293T cells: the 4-HNE-induced increase of Aβ40 level and Aβ40/42 ratio were both obviously inhibited in VPS35-depleted cells (Fig. 6k). No significant change of the Aβ42 levels was observed.

In HEK293T cells, (R)−4-HNE treatment induced early endosome fusion and enlarged Rab5+ puncta size (Supplementary Fig. 11b), indicating altered functionality of early endosomes. We then enriched early endosome fractions from WT HEK293T and 4-HNE-treated HEK293T cells using a commercial kit (Invent, ED-028) (Supplementary Fig. 11d) and measured the Aβ40 and Aβ42 levels. The data showed that (R)−4-HNE treatment led to elevation of both Aβ40 and Aβ42 levels in early endosomes than those untreated counterparts, with a decrease of Aβ40/42 ratio, but this decrease was not statistically significant (Supplementary Fig. 11e).

Therefore, the above results indicate that SORL1, as a linking receptor, can bind more (R)−4-HNE adducted APP, and facilitates more retromer-dependent retrieval of APP from early endosomes to the Golgi apparatus, and enhances the production of Aβ40 peptides.

### Lower ALDH2 activity ameliorates Aβ-induced microglial activation and phagocytic phenotype

Microglia are resident innate immune cells in the brain. It is generally believed that long-term, sustained microglia activation-mediated phagocytic uptake and secretion of inflammatory cytokines elicit neurotoxicity that is considered a vital driver of AD pathogenesis[40].

Microglia also cluster around Aβ plaques, leading to compact plaque microregions and preventing Aβ peptides from spreading, and undertake phagocytosis of plaques to clear them[41]. Here, we explored whether ALDH2 activity affects the function of microglia.

Previous studies have confirmed that lipopolysaccharide (LPS) and oligomeric Aβ40 (oAβ40) can induce microglial activation. Consistently, treatment of BV2 (mouse microglia) cells with LPS or oAβ40 for 24 h, but not fibrillar Aβ40 (fAβ40), resulted in activation, characterized by an enlarged cell body size (Fig. 7a, b) and significantly higher levels of interleukin (*Il*)−1β, *Il-6*, and tumor necrosis factor-alpha (*Tnf-α*) production (Fig. 7c, d). In contrast, oAβ40 failed to induce activation in Aldh2-knockdown BV2 cells (Supplementary Fig. 12a), in which case the cell size and levels of inflammatory cytokine production were similar to those observed in WT BV2 cells. Conversely, Aldh2 knockdown did not have any impact on LPS-induced microglia activation (Fig. 7a, b, d). Additionally, there were no significant differences in microglial morphology between *Aldh2*⁻/⁻ mice and control C57BL/6 mice (Supplementary Fig. 12b), in line with what was observed when comparing PBS-treated Aldh2-knockdown BV2 cells with their control (siNC) counterparts (Fig. 7a, b).

We evaluated the influence of Aldh2 activity on the ability of microglia to engulf latex beads as a measurement of phagocytotic capacity by using flow cytometry and fluorescence imaging. BV2 cells were treated with daidzin (20, 40, 60, 80, or 100 μM) to inhibit ALDH2 activity; CCK8 assay revealed decreased cell proliferation induced by high concentrations (Supplementary Fig. 12c) or long treatment times (Supplementary Fig. 12d). The data showed that Aldh2 knockdown, pharmacological inhibition (with daidzin), or activation (with the ALDH2 activator Alda-1), did not significantly affect the basal phagocytic function of BV2 cells (Supplementary Figs. 12e−i and 13a, b). oAβ40 markedly enhanced the phagocytic activity of WT BV2 cells toward latex beads; however, this effect was attenuated in Aldh2-knockdown BV2 cells (from 1.36-fold to 1.11- or 1.13-fold) (Fig. 7e, Supplementary Fig. 13c).

In AD, Aβ plaques are tightly enveloped by microglia. The microglia constitute a protective barrier around the Aβ plaques, and this barrier modulates the degree of plaque compaction and amyloid fibril surface area[38]. To investigate the relationship between rs671 polymorphism and the microglia barrier, immunofluorescence staining was conducted using Iba-1 and 6E10 antibodies to co-label microglia and Aβ plaques. These results confirmed the microglia barrier surrounding Aβ plaques in both APP/PS1 mice and in pathological AD. However, treatment of APP/PS1 mice with daidzin or possession of the *ALDH2* rs671 GA/AA genotype in pathological AD significantly decreased the recruitment of microglia in close proximity to Aβ plaques (Fig. 7f, g, Supplementary Fig. 12j, k).

Overall, these findings indicated that ALDH2 deficiency disrupted microglial functions in response to oAβ, including attenuated production of inflammatory mediators, as well as impaired phagocytosis and clustering around Aβ plaques.

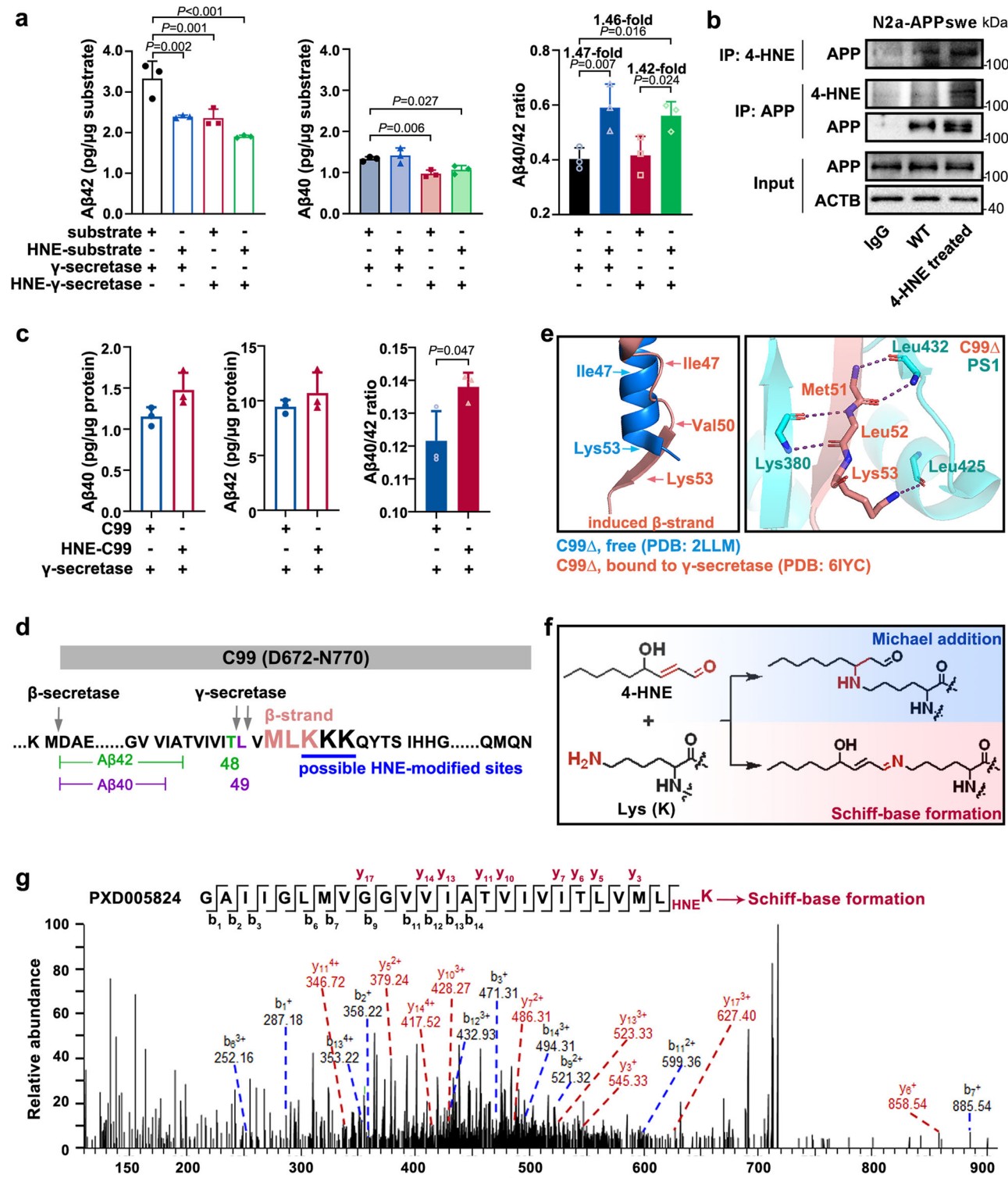

## Discussion

This study explored the relationship between *ALDH2* rs671 polymorphism and amyloid β pathology in 469 postmortem brains from the Human Brain Bank. Despite the existence of larger human brain tissue banks, this Human Brain Bank is the optimal platform for this study because of the high prevalence of the *ALDH2* rs671 variant only in East Asian populations (Chinese, Japanese, and Korean), which leads to a relatively high number of AD brain tissues with the GG, GA, and AA genotypes in this brain bank.

Low concentration of ethanol shows protective effects against Aβ toxicity in hippocampal neurons[42] and cardiac-cerebral vascular disease in GG genotype individuals[7,43]. Excessive ethanol exposure is detrimental to the brain and increases the risk for AD[42]. *ALDH2* rs671 G>A greatly reduces alcohol metabolism inducing toxic aldehyde load. Joshi et al. demonstrated worse neuropathology in the brains of *ALDH2* AA-allele mice relative to GG (wild-type) mice after chronic alcohol consumption[44]. Here, this study demonstrated that the *ALDH2* rs671 variant promotes Aβ pathology in postmortem brains with pathological AD in an ethanol-independent way, with increased amyloid

**Fig. 4 | (R)−4-HNE increases the Aβ40/42 ratio via covalent adduction on residue Lys53 of C99. a** Levels of human-derived Aβ40 and Aβ42 in in vitro γ-secretase cleavage assay. The natural and 4-HNE-modified substrates (HNE-substrate) were, respectively from N2a-APPswe cells, a mouse neuronal cell line stably over-expressing human APPswe, treated with PBS or 10 μM (R)−4-HNE for 4 h. Natural and 4-HNE modified γ-secretase complexes (HNE-γ-secretase) were extracted from HEK293T cells pretreated with PBS or 10 μM (R)−4-HNE for 4 h. n = 3 biologically independent samples. **b** APP or 4-HNE-adducted APP were immunoprecipitated from N2a-APPswe cells pretreated with PBS or 10 μM (R)−4-HNE for 4 h and probed with 6E10 or 4-HNE antibody, respectively. **c** Levels of Aβ40 and Aβ42 generated from recombinant C99 (with a C-terminal His-tag, expressed in Escherichia coli from pET28a) in in vitro γ-secretase cleavage assay. 4-HNE-modified C99 (HNE-C99) was generated from recombinant C99 by reaction with 50-fold (mol/mol) (R)−4-HNE for 1 h at 37 °C. n = 3 biologically independent samples. **d** Sequence of C99. The secondary β-strand structural elements of C99, and possible 4-HNE-modified sites, are indicated. **e** An overall view of a portion of the C99 transmembrane fragment (C99Δ) between the free stage (blue, PDB: 2LLM) and the γ-secretase-bound stage (pink, PDB: 6IYC) and a close-up view of the interaction of Met51−Lys53 fragment of C99 with PS1 through hydrogen bonds. **f** Potential reactions of the aldehyde and double bond groups in 4-HNE with proteins on −NH₂ group (Lys) via Michael addition and Schiff-base formation. **g** Representative MS/MS spectra of the peptide from C99 containing residue Lys53 modified by Schiff-base formation (Delta Mass, +138.10) with 4-HNE. Data are from deposited tandem mass-tag labeled proteomic data of amyloid plaques from postmortem brains with pathological AD and control (PXD005824). Data are presented as mean values ± SD. Statistical analysis was performed using a two-tailed Student's t-test for two groups (**c**) and one-way ANOVA with LSD post-hoc test for multiple groups (**a**). Source data are provided as a Source Data file.

plaque burden and a higher Aβ40/42 ratio. Autopsy studies established that Aβ plaque is a classic neuropathological hallmark of AD, accumulating progressively with age[45]. The current dominant view holds that Aβ plaques accumulate silently in the brain for decades before the clinical symptoms of AD become apparent[46]. The concept of "preclinical AD" indicates that individuals display neuropathological evidence of amyloid plaques and tau protein abnormalities but do not exhibit clinical dementia symptoms[47]. Our study found that ALDH2 rs671 G>A exacerbated Aβ pathology but did not conclude as a direct risk factor for the onset of AD. This aligns with the increasingly recognized complexity of the relationship between Aβ accumulation and the clinical manifestation of AD[48]. Besides, this study indicates that the change of Aβ pathology is ALDH2 enzyme activity-dependent. Accumulated 4-HNE, the major endogenous substrate of ALDH2, affects Aβ production. The (R)−4-HNE enantiomer adducts residue Lys53 of C99 via Schiff-base formation and alters C99 processing in favor of Aβ40 generation in the Golgi apparatus. Moreover, ALDH2 deficiency impairs microglia response to oAβ40, but not to LPS: ALDH2 deficiency decreases secretion of inflammatory mediators, the phagocytic capability of microglia, and their ability to surround and clear amyloid plaques.

Several types of research indicated the efficiency of the Aβ42/40 ratio in plasma and CSF to distinguish AD from a healthy population and to be a proxy for amyloid status in AD[18,49]. This study found that the ALDH2 rs671 variant altered the Aβ42/40 ratio, though with no association with AD risk. Given the high prevalence of ALDH2 rs671 A-variant in East Asians, it should be taken into consideration as an important correction factor when the Aβ42/40 ratio as a biomarker for AD screening in this population, thus in order to enhance AD diagnostic accuracy.

This study indicates the significant association of the ALDH2 rs671 variant with Aβ pathology for multiple reasons. On the one hand, the ALDH2 rs671 variant causes 4-HNE accumulation in the brain. Ando et al. first reported the immunoreactivity of 4-HNE in amyloid deposits[50]. Kelly et al. and our data show that 4-HNE can cause covalent cross-linking of Aβ peptides (Supplementary Fig. 14a), leading to more fibrillar aggregation and plaque deposition[51,52]. In human brains, there are two types of plaque, focal and diffuse plaques (Supplementary Fig. 14b). Liu et al. reported that most focal plaques are accompanied by microglial cells and correlate with aging and AD-related neuropathological changes[53]. This study revealed that the ALDH2 rs671 variant slightly increased the proportion of focal-type plaques in human brains (Supplementary Fig. 14c), which may be related to 4-HNE induced Aβ cross-linking making plaques more likely to aggregate into the focal-type. On the other hand, the ALDH2 rs671 variation leads to impaired phagocytic capability of microglia, thereby resulting in increased Aβ accumulation in brains. Moreover, activated microglia can wrap around plaques, leading to compact plaque microregions and preventing Aβ peptides from spreading. Here microglia clustering around plaques was found to decrease in postmortem brains with the

ALDH2 A-allele, which may accelerate the diffusion of Aβ "seeds", leading to increased spreading of amyloid plaques throughout different brain regions. Consistent with this, more plaque depositions were detected in deeper brain regions (Hipp, basal nuclei) in individuals with the GA/AA genotypes compared with those with the GG genotype.

As a crucial innate immune cell in the brain milieu, microglia act as a double-edged sword, exerting beneficial or detrimental influences[54]. On one hand, they secrete pro-inflammatory cytokines and neurotrophic factors to modulate neuronal survival and synaptic plasticity. On the other hand, they phagocytose debris and toxic proteins such as Aβ to prevent their accumulation and ensure homeostasis of the brain microenvironment. This study demonstrated that reduced ALDH2 activity disrupted both opposing effects of microglia. That is, reduced ALDH2 activity attenuated the production of inflammatory mediators in response to oAβ, thus reducing neuronal damage and neurotoxicity and showing a protective effect for neuronal cells; However, ALDH2 deficiency also impaired the Aβ clearance and microglia clustering around plaque deposits. As a result, this can exacerbate more Aβ accumulation and spread throughout whole brains[54,55].

Chirality is a critical issue in pharmaceutics. Due to stereospecific interactions of enantiomers, biological systems usually favor one enantiomer over the other. Enantiomers of a chiral compound can exhibit different activities, i.e., one enantiomer can have the desired effect while the other can be inactive or even toxic. For example, ibuprofen is a widely used chiral nonsteroidal anti-inflammatory drug; S-(+)-ibuprofen possesses the majority of the pharmacological activity, about 160 times more than the R-(−)-enantiomer[56]. Thalidomide is infamous for causing birth defects with phocomelia in about 10,000 newborns worldwide, related to the use of this drug in pregnant women. Later studies demonstrated severe embryotoxic and teratogenic effects of the S-enantiomer of thalidomide in rodent pups, while offspring of animals exposed to the R-form developed normally[57,58]. In the present study, we investigated the effect of the chirality of 4-HNE in Aβ generation. The data revealed that (R)−4-HNE promoted more production of shorter Aβ, while the S-form or the racemate did not affect Aβ length. Whether the (R)-enantiomer of 4-HNE plays a significant role in other disease pathologies warrants further investigation. In any case, despite the potential for lipid peroxidation to generate both enantiomers of 4-HNE (i.e., 4R and 4S), the resultant chirality markedly affects the corresponding molecular metabolism. Brichac et al. demonstrated that rat Aldh5A selectively oxidizes (R)-HNE rather than (S)-HNE in an Mg²⁺-dependent manner[59]. Glutathione (GSH) is an important compound for the detoxification of 4-HNE through either redox transformation of the aldehydic group or conjugation with the thiol group to generate GSHNE. Various glutathione S-transferase (GST) isoforms have been described to catalyze the generation of GSHNE. One of the GSTs, the A4-4 isoform, was reported to have a modest substrate preference for 4S-HNE[60]. Kinetic analysis combined with molecular modeling studies demonstrated a stereo-selectivity of the aldo−keto reductase AKR1B1 with a marked

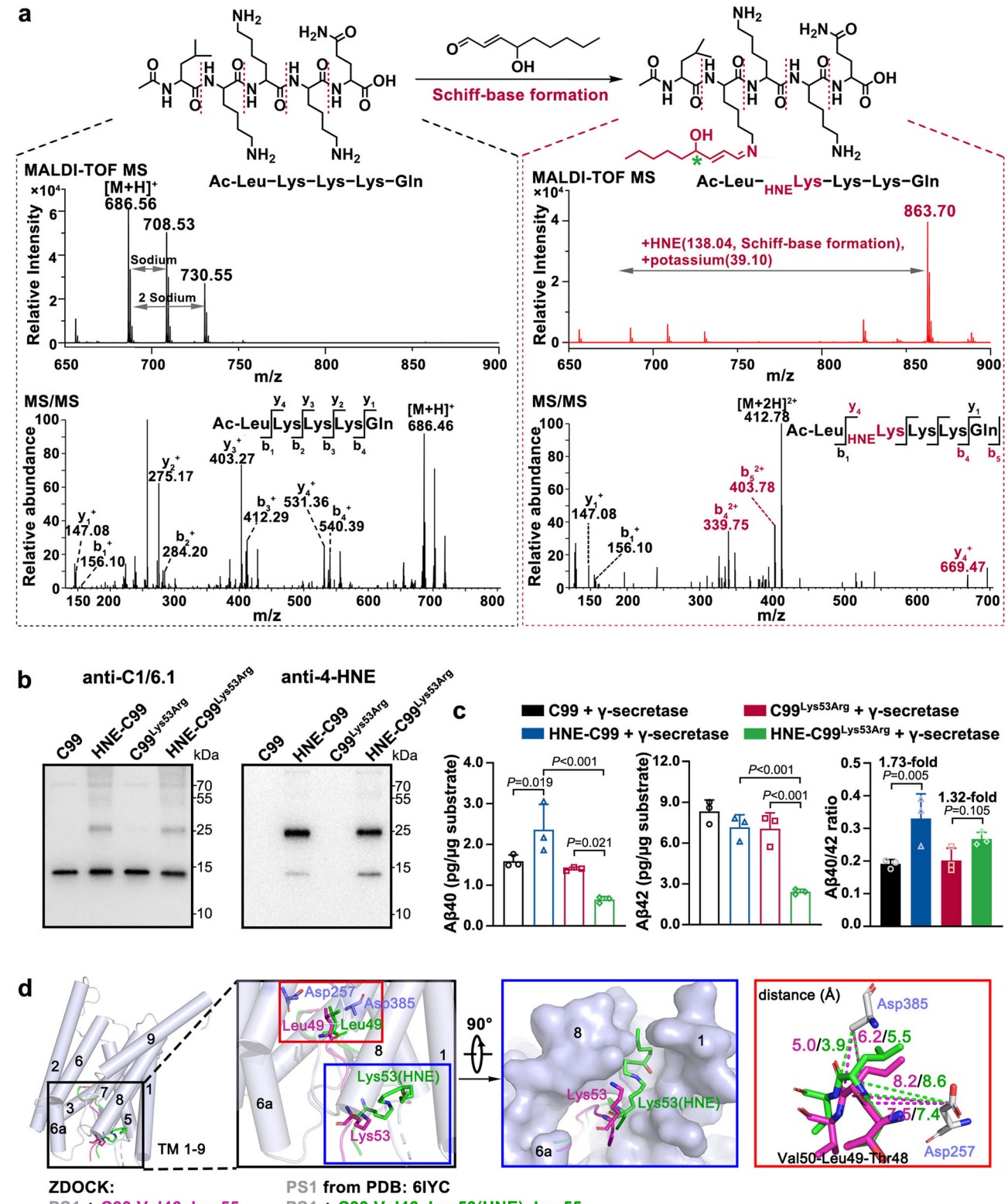

preference toward the 3S−4R-GSHNE diastereoisomer[61]. Thus, finding a stereoselective metabolic pathway of (R)−4-HNE would be important in AD. It is worth mentioning that the reaction of 4-HNE with −SH or −NH groups via Michael addition may generate another chiral center at the C3 atom, such as in reaction with GSH[62]. We observed 4-HNE adduction on the second Lys of peptide Ac-Leu–Lys–Lys–Lys–Gln via Michael addition (Supplementary Fig. 9f) as a minority product. This reactivity increases the complexity of research into the effects of

chirality of 4-HNE. The specific effects of (R)- and (S)−4-HNE in disease are poorly understood. The lack of tools to distinguish 4-HNE enantiomers, such as antibodies and functional imaging techniques for studying chiral 4-HNE in vivo, hampers this research, as does the fact that only a precursor of (R)−4-HNE is commercially available.

4-HNE is a small molecule that can easily diffuse throughout the whole cell body. Despite adduction on APP, the highly reactive 4-HNE also adducts with Cys, His, and Lys residues of other proteins,

**Fig. 5 | (*R*)−4-HNE adducted on Lys53 by Schiff-base formation caused positional deviation of C99, leading to increased Aβ40/42 ratio. a** The Schiff-base formation reaction of Ac-Leu-Lys–Lys–Lys–Gln (residues 52-56 of C99) with (*R*)-4-HNE. Matrix-assisted laser desorption/ionization-time of flight MS (MALDI-TOF MS) detection of natural peptide Ac-Leu-Lys–Lys–Lys–Gln (left, upper) and 4-HNE-modified peptides (right, upper). Representative MS/MS spectra of natural peptide Ac-Leu-Lys–Lys–Lys–Gln (left, lower) and modified peptide (right, lower) with 4-HNE adduction on the first Lys residue (Lys53 of C99) via Schiff-base formation, the main product of this reaction. **b** WB verification of recombinant C99, Lys53Arg mutant C99 (C99$^{Lys53Arg}$), and 4-HNE-modified HNE-C99 and HNE-C99$^{Lys53Arg}$. C99 was detected by C1/6.1 antibody. 4-HNE-modified proteins were detected by 4-HNE antibody. **c** Levels of generated Aβ40 and Aβ42 in in vitro γ-secretase cleavage assays from recombinant C99, C99$^{Lys53Arg}$, and their 4-HNE-modified forms,

determined by ELISA assay. $n = 3$ biologically independent samples. **d** The predicted docking structures of human γ-secretase with peptide Val46–Lys55 of C99 (purple) or with modified peptide Val46–Lys53(HNE)–Lys55 (green) by ZDOCK and sequence alignment was done in PyMOL. The structures of PS1 (the main γ-secretase component) and natural Val46–Lys55 peptide were derived from the Cryo-EM structure (PDB: 6IYC). The 4-HNE modification on Lys53 (after Schiff-base formation) was manually added in PyMOL based on the natural peptide structure from 6IYC. The dashed lines and the numbers represent the distance between Asp385 or Asp257 of PS1 and −NH or C=O groups of residue Leu49 of C99 [purple, natural Val46-Lys53; green, modified Val46-Lys53(HNE)-Lys55]. Data are presented as mean values ± SD. Statistical analysis was performed using one-way ANOVA with LSD post-hoc test for multiple groups. Source data are provided as a Source Data file.

modulating or disrupting the function of these macromolecules. For example, 4-HNE adduction on the proteasome complex can affect the efficiency of protein degradation. We found that inhibition of ALDH2 activity did not change the mRNA level of *PS1* but increased PS1 and total tau expression levels (Supplementary Fig. 5b, d, f, h, l). Thus, changes in ALDH2 expression may lead to changes in non-enzyme-dependent signaling pathways.

Several small molecule activators have been reported to improve the activity of mutated ALDH2, represented by Alda-1. Yang et al. found that Aldh2 overexpression improved cognitive function of APP/PS1 mice[63]. *ALHD2* rs671 G>A greatly increased aldehydic load and exacerbated ethanol-induced neuropathology change. In *Aldh2* AA-homozygous knock-in mice, Alda-1 administration significantly blunted the ethanol-induced increases in Aβ and neuroinflammation in vivo and in primary neurons and astrocytes, with the underlying mechanism of effective clearance of ethanol-derived acetaldehyde by Alda-1[44]. Without ethanol consumption, we found Alda-1 administration also slightly reduced the plaque deposition area and mildly decreased the levels of Aβ40 and Aβ42 in APP/PS1 mice, with no significance, maybe due to poor delivery efficiency of Alda-1 (Supplementary Fig. 15a, b). In vitro, Alda-1 did not significantly affect the wild-type neuron with no significant altering of protein expression profiling in N2a-APPswe (Supplementary Fig. 15c). Wild-type microglia BV2 showed inhibited proliferation (Supplementary Fig. 15d, e) but unaffected phagocytic ability (Supplementary Fig. 12h) after treatment of 20 μM or more Alda-1. The altered metabolisms of lipids or steroids (Supplementary Fig. 15f–h, Supplementary Data 3) may account for the inhibition effect induced by Alda-1.

There are some limitations in this study. First, more individuals with the rs671 AA genotype are required. The overall AA prevalence was quite low in the dataset, only 3.0%. Among the total 14 AA carriers enrolled in this research, a high incidence (8 individuals, 57.1%) of pathological AD was found. However, the number of AA individuals is insufficient to determine whether AA is a risk factor for AD. Second, we found that 4-HNE modified residue Lys53 of C99 via Schiff-base formation, which disrupted the key β-sheet structure and caused a slight positional shift in the binding of C99 to γ-secretase due to steric hindrance. However, after ZDOCK prediction, the RosettaDock analysis faces limitations in docking large and flexible protein complexes with more than 1500 amino acids or those with individualized modifications, such as 4-HNE modification. New algorithms or crystal structures are required to precisely determine the mechanism of 4-HNE-induced Aβ alteration.

In summary, this study explored the correlation between *ALDH2* rs671 polymorphism and postmortem AD pathology grade, using the largest number of human brains with rs671 A-variant in East Asian population (Fig. 8). We demonstrated the high risk of rs671 A-variant for Aβ pathology, with more plaque deposits and higher Aβ40/42 ratio generated in neurons, and revealed that the chiral (*R*)−4-HNE adducts on Lys53 of C99 via Schiff-base formation thus altering C99 process to favor Aβ40 generation over Aβ42 in the Golgi apparatus. Also, reduced

ALDH2 activity impaired microglial functions in pathological AD brains, both lowered inflammatory factor secretion, as well as Aβ phagocytosis and spreading. These findings indicated that a positive correlation between *ALDH2* rs671 polymorphism and Aβ pathology should be considered when applying antibodies for Aβ plaques or oligomers clearance or when the ratio of Aβ peptide segments as a diagnostic marker for AD in East Asia.

## Methods

### Ethics
All experimental protocols described in this study were approved by the Ethics Committee of the Institute of Basic Medical Sciences Chinese Academy of Medical Sciences and complied with all ethical regulations.

### Human samples
Postmortem human brain samples were provided by the National Human Brain Bank for Development and Function, affiliated with the Chinese Academy of Medical Sciences and Peking Union Medical College, located in Beijing, China. Individuals were eligible for inclusion based on the following criteria: (1) a definitive pathological diagnosis of Alzheimer's disease (AD) or control (CTRL); (2) exclusion of individuals with a history of long-term heavy alcohol consumption; (3) absence of other CNS disorders except AD; and (4) confirmation of *ALDH2* rs671 or *ADH1B* rs1229984 genetic sequencing outcomes. Ultimately, 469 individuals (267 males and 202 females) were included in the *ALDH2* rs671 analysis, while 190 individuals (107 males and 83 females) were enrolled for *ADH1B* rs1229984 sequencing.

The human brain tissues were initially preserved at −80 °C and subsequently fixed with 4% paraformaldehyde (PFA) and cryoprotected in 30% sucrose overnight before sectioning. All procedures involving human tissue were conducted following the Standardized Operational Protocol for Human Brain Banking in China[64].

### Animal experimental models and drug treatment
Twelve male APP/PS1 transgenic mice (see the Glossary in Supplementary Notes) were used in this study. These mice (with the C57BL/6 genetic background) were from JKbiot (Nanjing Junke Bioengineering Co., Ltd., China). *Aldh2*$^{-/-}$ mice ($n = 5$, male, 3-month-old, strain no. T013848), lacking the aldehyde dehydrogenase 2 gene, were from GemPharmatech (Nanjing, China). All mice were maintained on a reversed 12 h:12 h light:dark cycle at constant temperature (22 ± 1 °C) and humidity with free access to food and water.

### Drug treatment and brain tissue preparation
Alda-1 was dissolved in 50% DMSO/50% PEG400 (v/v). Daidzin was dissolved in 10% DMSO/40% PEG400/5% Tween-80/45% saline (v/v/v/v). Four 2-month-old male APP/PS1 mice were administered physiological saline via intragastric infusion (*iG*) as the control group,

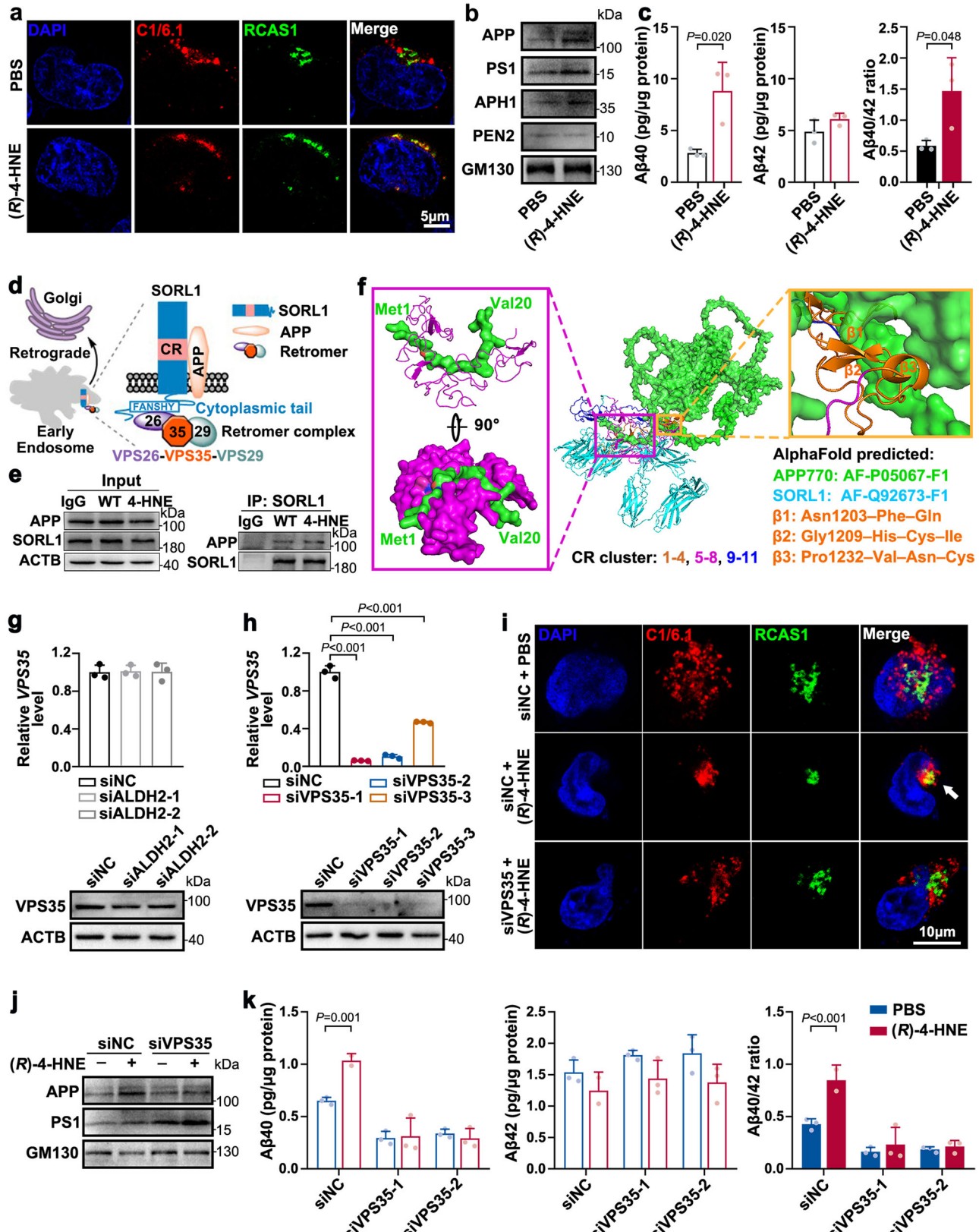

while four mice were given daidzin (at a dose of 150 mg/kg/day of body weight) or four for Alda-1 (at a dose of 15 mg/kg/day of body weight) via *iG* for 2 months. Because of the impact of the COVID-19 pandemic, no treatment was administered to the mice during the period from 5 to 7 months of age. At 7 months of age, all mice were deeply anesthetized with pentobarbital sodium and intracardially

perfused with ice-cold PBS. The left-brain tissues were stored at −80 °C for WB and ELISA. The right-brain tissues were fixed in 4% PFA, followed by gradient dehydration in 20–30% sucrose solution and embedding in OCT compound, and stored at −80 °C for subsequent immunohistochemistry and immunofluorescence analyses.

**Fig. 6 | (R)−4-HNE-adducted APP retrogrades to the Golgi apparatus aggravating Aβ40 generation. a** Representative high-magnification single-plane confocal images of HEK293T cells stained with C1/6.1 (red, APP) and RCAS1 (green, Golgi apparatus). HEK293T cells were pretreated with PBS or 2 μM (R)−4-HNE for 24 h. Scale bar, 5 μm. **b** and **c** Golgi apparatus was isolated from HEK293T cells pretreated with PBS or (R)−4-HNE by using a Golgi Apparatus Enrichment Kit (Invent, GO-037). **b** WB of APP and γ-secretase components, and **c** levels of Aβ40 and Aβ42 in PBS- or (R)−4-HNE-treated Golgi apparatus lysates. $n = 3$ biologically independent samples. **d** Scheme of SORL1 as a molecular link for retromer-dependent sorting of APP[34–36]. Image was created using elements from ChemBio-Draw Ultra software 14.0. **e** Immunoprecipitation of SORL1 and natural or 4-HNE modified APP, without or with 4-HNE treatment. Two independent biological replicates were performed. **f** The calculated protein-protein docking results of APP770 and SORL1 domains (YWTDs−CR cluster−FN repeats) by Rosetta software. The structures of APP770 and SORL1 are predicted by AlphaFold downloaded from UniProt (for APP770: AF-P05067-F1, for SORL1: AF-Q92673-F1). The CR cluster of SORL1 contains 11 complement-type repeat domains and is indicated as orange (CR 1–4), purple (CR 5-8), and blue (CR 9-11). **g** Relative mRNA and protein levels of VPS35 after ALDH2 knockdown by RNA silencing in HEK293T cells. $n = 3$ biologically independent samples. **h−k** VPS35 levels were suppressed by RNA silencing in HEK293T cells, and cells were then treated with PBS or 2 μM (R)−4-HNE for 24 h. **h** Verification of VPS35 knockdown by RT-qPCR and WB. $n = 3$ biologically independent samples. **i** Double staining with C1/6.1 (red, APP) and RCAS1 (green, Golgi apparatus) antibodies in the above cells. Scale bar, 10 μm. **j** WB of APP and PS1 in isolated Golgi apparatus lysates. **k** Levels of Aβ40 and Aβ42 in Golgi apparatus lysates. siNC, negative control small interfering RNA. $n = 3$ biologically independent samples. Data are presented as mean values ± SD. Statistical analysis was performed using a two-tailed Student's t-test (**c**, **g**, **h**, **k**). Source data are provided as a Source Data file.

## Cell culture

Human neuroblastoma SH-SY5Y cells were cultured in Dulbecco's Modified Eagle's Medium (DMEM) and Ham's F12 (HyClone, USA), supplemented with 10% fetal bovine serum (FBS; Gibco, Australia). Human embryonic kidney (HEK293T) and mouse microglial BV2 cells were cultured in DMEM supplemented with 10% FBS. The cell lines were obtained from the Cell Resource Center, Peking Union Medical College (SH-SY5Y, #PUMC000291; HEK293T, #PUMC000091; BV2, #PUMC000063). Mouse neuroblastoma N2a-APPswe cells, which have been transfected with the human APP gene with the Swedish mutation, were kindly gifted by Prof. B.H. Xu (Shenzhen Center of Disease Control and Prevention, China), and were grown in DMEM supplemented with 10% FBS and 200 μg/mL geneticin. All cell cultures were maintained at 37 °C in a humidified atmosphere with 5% $CO_2$. Cell passage was performed approximately every 2–3 days.

## Small molecules and biomolecules

(R)−4-hydroxynonenal dimethylacetal (CAS No. 119009-01-7, Catalog No. 870608H, Avanti, purity >99%) was obtained from Merck. Racemic 4-HNE ((R/S)−4-Hydroxy-2E-nonenal, CAS No. 75899-68-2, Catalog No. HF0079, purity >99%) was obtained from ChemHi-Future. Alda-1 (CAS No. 349438-38-6, Catalog No. A15805, purity >98%) and daidzin (CAS No. 552-66-9, Catalog No. A10283, purity >98%) were obtained from AdooQ BioScience. The peptides Aβ1−40 (Catalog No. 051516, purity >95%), Ac-Leu-Lys-Lys-Lys-Gln (customized, purity >93%), were obtained from GL Biochem (Shanghai) Ltd.

## Antibodies

Anti-ALDH2 (Proteintech, 15310-1-AP, 1:4000), anti-4-HNE (Abcam, ab46545, 1:2000), anti-APP C-Terminal Fragment (C1/6.1, Biolegend, 802801, 1:500), anti-β-amyloid (1−16) (6E10, Biolegend, 803001, 1:1000), anti-β-amyloid (1−40) (Cell Signaling Technology, 12990, 1:1000), anti-β-amyloid (1−42) (Cell Signaling Technology, 14974, 1:1000), anti-GM130 (Cell Signaling Technology, 12480S, 1:1000), anti-TGN46 (Invitrogen, MA3-063, 1:1000), anti-presenilin 1 (Sigma, MAB5232, 1:500), anti-PEN2 (Abcam, ab18189, 1:500), anti-APH1 (Invitrogen, PA1-2010, 1:1000), anti-Rab5 (Cell Signaling Technology, 3547, 1:1000), anti-RCAS1 (Cell Signaling Technology, 12290S, 1:1000), anti-VDAC (Cell Signaling Technology, 4661S, 1:1000), anti-LAMP2 (Proteintech, 27823-1-AP, 1:1000), anti-CANX (Cell Signaling Technology, 2679, 1:1000), anti-VPS35 (Abcam, ab10099, 1:1000), anti-LC3A (Cell Signaling Technology, 4599, 1:1000), anti-Iba1 (Wako, 019-19741, 1:500), anti-tau (Cell Signaling Technology, 46687, 1:1000), anti-phospho-S396-tau (Abcam, ab32057, 1:1000), anti-α-tubulin (GeneTex, GTX628802, 1:10,000), anti-β-actin (GeneTex, GTX124213, 1:10,000), anti-GAPDH (MBL, M171-3, 1:10,000), and anti-SorLA/SORL1 antibody (Abcam, ab190684, 1:1000).

## Proteomic analysis

**Protein extraction and TMT labeling.** Quantitative proteomic analysis using the TMT strategy, as previously described[65], was carried out in this study. Human brain tissues, or N2a-APPswe, or BV2 cell pellets, were dissected and homogenized immediately in 8 M urea buffer. The protein concentration was quantified using a NanoDrop 2000 spectrophotometer, and 100 μg of total protein in each sample was used for subsequent analysis. The sample was reduced with 10 mM dithiothreitol at 37 °C for 30 min and alkylated with 25 mM iodoacetamide for 30 min at room temperature in the dark. The protein extracts were then digested overnight at 37 °C with endopeptidase Trypsin/Lys-C (4 μg). After digestion, the protein digests were desalted and placed in 50 μL 200 mM triethylammonium bicarbonate buffer, followed by labeling with 0.8 mg TMT reagents at room temperature for 1 h.

N2a-APPswe cells treated with daidzin, Alda-1, and PBS were labeled with TMTs as follows: TMT-128 for PBS-treated cells; TMT-130 for Alda-1-treated cells; and TMT-131 for daidzin-treated cells.

BV2 cells treated with Alda-1 and PBS were labeled with TMTs as follows: TMT-128 for PBS-treated cells, and TMT-130 for Alda-1-treated cells.

Six SETs of human brain tissues were labeled with TMTs.

**Six SETs of proteomics on human brain tissue**

|      | TMT-126 | TMT-127 | TMT-128 | TMT-129 | TMT-130 |
|------|---------|---------|---------|---------|---------|
| SET1 | AD2     | AD8     | AD9     | CTRL7   | CTRL12  |
| SET2 | AD1     | AD7     | AD5     | CTRL3   | CTRL2   |
| SET3 | AD6     | AD10    | AD3     | CTRL9   | CTRL8   |
| SET4 | AD4     |         |         | CTRL6   | CTRL4   |
| SET5 |         | CTRL12  | CTRL13  | CTRL11  | CTRL5   |
| SET6 | CTRL10  |         | CTRL14  |         |         |

Following the labeling reactions, 5 μL 5% hydroxylamine was added, and the mixtures were incubated for 20 min to stop the reactions. All of the labeled extracts within each SET were mixed, desalted, dried, and then dissolved in 20 μL 0.1% formic acid.

**HPLC fractionation.** The mixture was then fractionated on a C18 column (XBridge, 4.6 × 250 mm, 5 μm, 300 Å, Waters) at a flow rate of 1.0 mL/min. The peptides were separated using a gradient of mobile phase B (mobile phase A: $H_2O$ with ammonium hydroxide, pH 10; mobile phase B: 98% acetonitrile with ammonium hydroxide, pH 10), with varying percentages of phase B applied at different time intervals. Specifically, the gradient started with 5–8% phase B from 0 to 5 min, followed by 8–18% phase B from 5 to 40 min, 18–32% phase B from 40 to 62 min, 32–95% phase B from 62 to 64 min, and 95% phase B from 64 to 72 min. The resulting 47 fractions were collected at 1.5-min intervals,

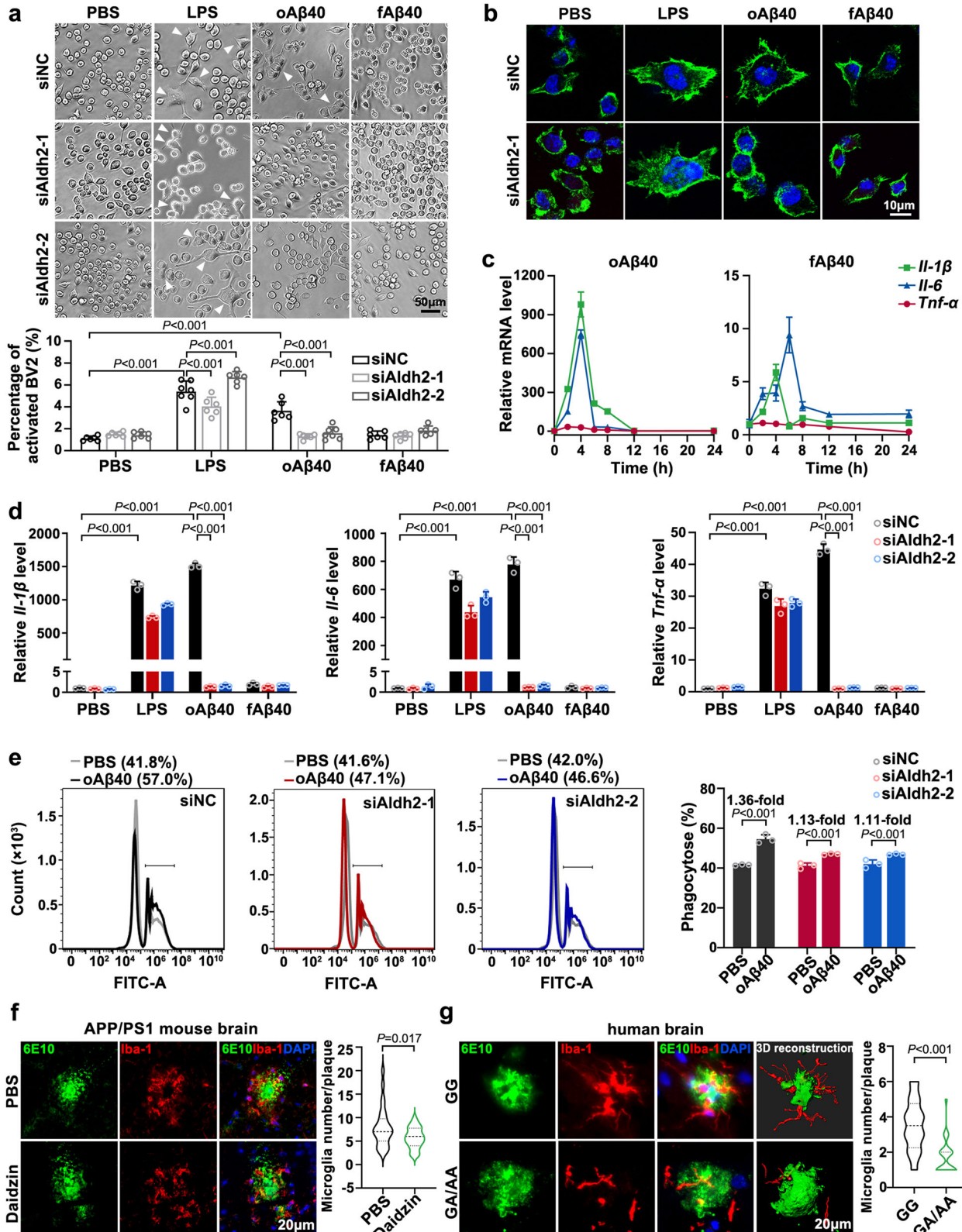

ranging from 2 to 72 min, dried in a vacuum concentrator, and combined. Finally, the combined fractions were dissolved in 20 µL 0.1% trifluoroacetic acid for liquid chromatography (LC)–MS/MS analysis.

**LC–MS/MS analysis.** To perform LC–MS/MS analysis in this study, an Orbitrap Fusion Tribrid Mass spectrometer (Thermo Fisher Scientific, MA, USA) was employed. The protein digests were separated using an

UltiMate 3000 RSLCano System (Thermo Fisher Scientific, MA, USA) with a flow rate of 0.3 µL/min over a 108-min gradient elution (0–8 min, 4% B; 8–11 min, 4–10%; 11–88 min, 10–25%; 88–98 min, 25–50% B; 98–108 min, 50–99%). The LC mobile phase A is 0.1% FA in water and phase B is 0.08% FA in ACN. The analytical column employed in the separation process consisted of a fused silica capillary (75 µm × 150 mm; Upchurch, Oak Harbor, WA, USA) packed with C18

**Fig. 7 | Lower ALDH2 activity ameliorated Aβ-induced microglial activation and phagocytic phenotype. a** and **b** Representative images of BV2 cells with Aldh2 silencing, followed by treatment with lipopolysaccharide (LPS) (1 μg/mL), oligomeric Aβ40 (oAβ40, 1 μM), or fibrillar Aβ40 (fAβ40, 1 μM) for 24 h. **a** Bright-field images and quantification of activated BV2 cells. Scale bar, 50 μm. $n = 6$ randomly selected fields per group. **b** Fluorescence microscopy of activated BV2 cells labeled with Alexa Fluor 488 phalloidin (green). Scale bar, 10 μm. **c** Relative *Il–1β*, *Il-6*, and *Tnf-α* mRNA levels in wild-type BV2 cells treated with oAβ40 (1 μM) or fAβ42 (1 μM) for 0, 2, 4, 6, 8, and 12 h, respectively. $n = 3$ biologically independent samples. **d** Relative *Il-1β*, *Il-6*, and *Tnf-α* mRNA levels in BV2 cells with Aldh2 knockdown and treatment with LPS (100 ng/mL), oAβ40 (1 μM), or fAβ42 (1 μM) for 4 h. $n = 3$. **e** Flow cytometry histogram showing the phagocytosis rate of BV2 cells with Aldh2 knockdown, stimulated with oAβ40 (1 μM) for 24 h and then co-incubated with latex beads for 4 h. $n = 3$ biologically independent samples. The flow-cytometry gating strategies were shown in Supplementary Fig. 13c. **f** Representative images of Aβ plaques (6E10, green) and microglia (Iba-1, red) in the cortex from APP/PS1 mice and daidzin-treated APP/PS1 mice. Quantification of microglial cells within 20 μm of a plaque ($n = 28$ plaques from 3 mice per group). Scale bar, 20 μm. **g** Representative images of Aβ plaques (6E10, green) and microglia (Iba-1, red) in the prefrontal lobe from pathological AD with rs671 GG genotype ($n = 3$) or the GA/AA genotype ($n = 3$). Quantification of microglial cells within 20 μm of a plaque ($n = 24–28$ plaques from 3 brains with AD per group). Scale bar, 20 μm. siNC negative control small interfering RNA. Data are presented as mean values ± SD. Statistical analysis was performed using one-way ANOVA with LSD post-hoc test for multiple groups (**a**, **d**) or two-tailed Student's *t*-test for two groups (**e**–**g**). Source data are provided as a Source Data file.

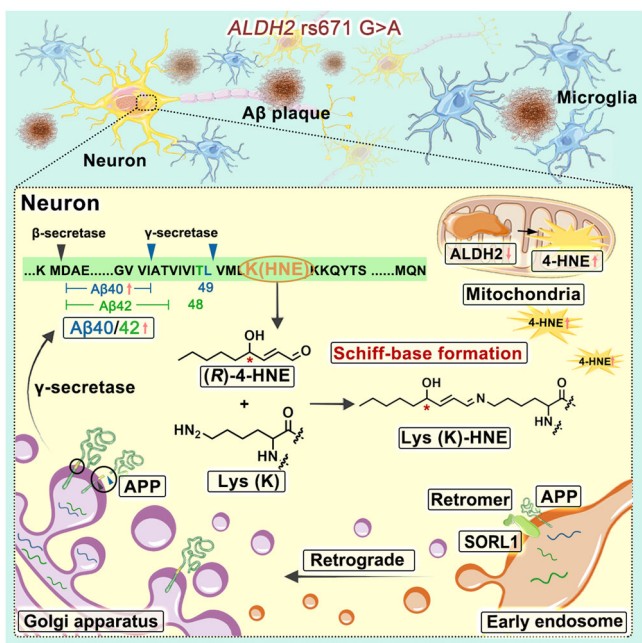

**Fig. 8 | Graphical abstract summarizing the results.** Image was created using Photoshop CS6 with Servier Medical Art elements (https://smart.servier.com).

resin (5 μm, 300 Å; Varian Lexington, MA, USA). Xcalibur 2.1.2 software was used to configure the mass spectrometer in data-dependent acquisition mode. Specifically, a single full-scan mass spectrum was acquired using the Orbitrap (350–1550 *m/z*, 120,000 resolutions), which was followed by 3-s data-dependent MS/MS scans at 35% normalized collision energy, utilizing higher energy C-trap dissociation. For TMT-labeled proteomics detection in this study, 4 or 12 samples were analyzed in each SET.

**Protein identification and bioinformatic analyses.** The Proteome Discoverer 2.4 software package (Thermo Fisher Scientific, MA, USA) was used to analyze MS/MS spectra. Spectra were searched against the human or mouse proteome from the UniProt Swiss-Prot database (human, released March 13, 2023; mouse, released November 10, 2023). Trypsin was selected as the proteolytic enzyme, and two missed cleavage sites were allowed. Signal to noise threshold was set at 1.5. The precursor mass tolerance was 10 ppm, and the fragment mass tolerance was 0.02 Da. Carbamidomethylation of cysteine, TMT6plex modification of lysine, and peptide *N*-terminus were set as the fixed modification. The oxidation of methionine and acetylation of the protein *N*-terminus were set as the variable modifications. As a filter, the minimum peptide length was 6, a 1% false-positive rate at the protein level and at the peptide level was set, and each protein contained at least one peptide.

Volcano plot was used to screen differentially expressed proteins. Here, we used a two-tailed Student's *t*-test to compare the differential expression of proteins between two groups for the TMT-labeling quantitative proteomics data, followed by Benjamini–Hochberg multiple testing correction to calculate the adjusted *P*-value. The threshold for differentially expressed proteins screening was set as adjusted *P* value ≥ 0.05, fold change ≤ 0.83 or ≥1.20. WikiPathways enrichment analysis was performed using the WebGestalt online toolkit (https://www.webgestalt.org/) and visualized in a bubble plot.

### Tissue processing for immunohistochemistry

For the study of human samples, paraffin-embedded human brain samples were sectioned at a thickness of 5 μm. Sections were deparaffinized in xylene, followed by rehydration in decreasing grades of alcohol. To retrieve antigens via heat induction, sections were then transferred to 0.01 M citrate buffer (pH 6.0) and heated at 95 °C for 10 min, then cooled to room temperature. After incubating in 3% $H_2O_2$ for 30 min in a humid environment, sections were incubated overnight at 4 °C with primary antibody.

For the study of mice, the brains of deeply anesthetized mice were intracardially perfused with PBS (25 mL) and then extracted from the cranium. The brain samples were fixed in 4% PFA at 4 °C for 24 h, followed by immersion in 20% and 30% sucrose solutions for 72 h each, before being embedded in the OCT compound. Fresh cryosections (10–12 μm) were prepared and treated with 0.3% Triton X-100 for 30 min to increase permeability and with 3% $H_2O_2$ for 30 min to block endogenous peroxidase activity. The sections were then incubated with primary antibody overnight at 4 °C in a humid chamber.

Subsequently, signal amplification was performed using the horseradish peroxidase-diaminobenzidine (HRP-DAB) Two-Step IHC Detection Kit (ZSGB-BIO, PV-9001, PV-9002). The sections were subjected to gradient dehydration and sealed with a neutral resin before images were captured using a Leica DM6B microscope (Leica Microsystems, Germany). The quantification of staining was conducted in a blinded fashion using Image Pro Plus software.

### Immunofluorescence

*For human brain tissue*: Human brain sections at a thickness of 10 μm were from frozen human brains embedded in OCT compound. Antigen retrieval was performed by immersing the sections sequentially in citrate buffer (pH 6.0) at 95 °C for 15 min, followed by cooling to room temperature. The sections were then treated with 5% bovine serum albumin (BSA) for 1 h and subsequently incubated overnight at 4 °C with primary antibodies, including anti-Iba1 (1:200) and anti-6E10 (1:200).

*For mouse brain tissue*: Frozen mouse brain sections at a thickness of 12 μm were treated with 5% BSA at room temperature for 1 h,

followed by overnight incubation at 4 °C with primary antibodies, including anti-Iba1 (1:500) and anti-6E10 (1:500).

*For cells*: Cells were cultured on coverslips and fixed for 30 min in 4% PFA. Before immunostaining, the sections were washed three times with 0.01 M PBS and treated with 0.3% Triton X-100 for 30 min at room temperature. Then, the cells were blocked with 5% BSA for 30 min at room temperature and incubated with primary antibody in 5% BSA at 4 °C overnight. The following primary antibodies were used: anti−4-HNE (1:200), anti-RCAS1 (1:100), anti-C1/6.1 (1:200), anti-Lamp2b (1:100), anti-Rab5 (1:150), and anti-CANX (1:100).

After primary antibody incubation, the sections were incubated with Alexa 488- or 594-conjugated secondary antibodies for 1 h at room temperature. Only human tissue sections were then treated with 0.1% Sudan Black for 15 min in the dark to decrease background staining. Excess Sudan Black was removed by rinsing with 70% alcohol. Nuclei were stained with 4′,6-diamidino-2-phenylindole (DAPI) fluorescence signals were detected using a Leica DMi8 microscope or a Leica TCS SP8 gated STED microscope (Leica Microsystems).

## Western blot (WB)

Collected cells or frozen human brain tissues were blended in RIPA lysis buffer (Solarbio, R10010) supplemented with a protease inhibitor cocktail (Roche, 04693132001). A sonication step was performed before centrifugation at 12,000×$g$ for 15 min at 4 °C. The supernatants were collected. Proteins were loaded onto SDS−polyacrylamide gels according to BCA quantification, separated, and transferred onto polyvinylidene fluoride membranes by wet electrophoretic transfer. The membranes were blocked for 1 h at room temperature with rocking in 5% nonfat milk in TBST (400 mM Tris−HCl, 3 M NaCl, and 1% Tween-20, pH 7.4). After blocking, the membranes were incubated with primary antibody overnight at 4 °C. The secondary antibody, either anti-mouse or anti-rabbit, was conjugated to HRP and was applied for 1 h at room temperature. Protein bands were visualized using chemiluminescence, and protein detection was performed using the ChemiDoc XRS chemiluminescent gel imaging system. Image Lab software (Bio-Rad Laboratories) was used for analyses. To ensure equal sample loading in each lane, β-actin or GAPDH was used for normalization. The uncropped gels and blots were shown in the Source Data.

## ELISA

Protein lysates from brain tissues or cells were obtained using RIPA buffer supplemented with a protease inhibitor, followed by centrifugation at 12,000×$g$ for 15 min to extract the proteins. The levels of Aβ42 and Aβ40 peptides were measured using a specific human amyloid-β peptide 1−42 ELISA Kit (Cusabio, CSB-E10684h) and a human amyloid-β peptide 1−40 ELISA Kit (Cusabio, CSB-E08299h), respectively. The experimental procedure was performed in accordance with the manufacturer's instructions.

## Small interfering RNA transfection

Control and specific siRNAs from RiboBio (Guangzhou, China) were used to perform transient knockdown experiments. The specific siRNA sequences are shown in Supplementary Table 8.

Cells were seeded in six-well plates at 60−80% confluence and transfected the next day with 10 nM siRNA oligonucleotide by using Lipofectamine™ RNAiMAX Reagent (Invitrogen, 13778-150) according to the manufacturer's instructions. The cells were harvested 48 h later for RT-qPCR, ELISA, or WB.

## ALDH2 activity assay

The enzymatic activity of ALDH2 was analyzed using the colorimetric mitochondrial aldehyde dehydrogenase (ALDH2) Activity Assay Kit (Abcam, ab115348, Cambridge, MA) according to the manufacturer's recommendations using 600 µg of total protein from cell samples.

## Co-immunoprecipition

WT substrate and 4-HNE-adducted substrate were obtained using a NE-PER Nuclear and Cytoplasmic Extraction Reagents Kit (Thermo, 78833) from N2a-APPswe cells pretreated without or with 10 µM (*R*)-4-HNE for 4 h. Nuclei were discarded. Membrane protein (500 µg) extracted from N2a-APPswe cells was prepared for immunoprecipitation assays. The target proteins were immunoprecipitated using primary antibodies (2 µg) through incubation at 4 °C for 1.5 h, followed by capture using Protein A/G PLUS-Agarose beads (Santa Cruz Biotechnology, sc-2003) (20 µL) overnight at 4 °C. The immunoprecipitates were then washed five times with RIPA buffer with centrifugation at 1000×$g$ for 5 min at 4 °C. After magnetic separation, the beads were mixed with loading buffer and boiled at 100 °C for 3 min to elute proteins. The eluted samples were subsequently analyzed by WB.

## Preparation of (*R*)−4-HNE and (*S*)−4-HNE

Standard (*R*)−4-HNE was obtained from its stable precursor (*R*)−4-hydroxynonenal dimethylacetal (Avanti, 870608). The precursor was deprotected by incubation with 1 mM cold HCl at 4 °C for 1 h. Fresh standard 4-HNE was prepared and diluted in a suitable medium before experiments.

Chiral separation of (*R*)−4-HNE and (*S*)−4-HNE from (±)−4-HNE was performed by using a Shimadzu LC-20AD HPLC system. Samples were analyzed and separated on a chiral column (4.6 × 150 mm, CHIRALPAK AS-H) using 95% hexane and 5% isopropanol as the mobile phase at a flow rate of 1.0 mL/min. After evaporating solvents, purity was determined by chiral HPLC. The products were stored at −80 °C.

## Preparation of recombinant C99 and C99$^{Lys53Arg}$

For preparation of recombinant C99 (pET28a, *C*-terminal His-tag) or recombinant C99$^{Lys53Arg}$ (pET28a, *C*-terminal His-tag), transformed *Escherichia coli* BL21 (DE3) cells were grown at 37 °C to OD$_{600 nm}$ ~ 0.6 and induced with 0.2 mM isopropyl-β-D-thiogalactopyranoside. After 16 h of induction at 22 °C, the cells were collected, resuspended in buffer containing 200 mM Tris, pH 8.0, 500 mM NaCl, 5 mM imidazole, and 5% glycerol, and disrupted by sonication for 9 min. Following centrifugation at 8000×$g$ for 20 min, the supernatant was passed through a 0.45-µm pore-size filter and loaded onto a HisTrap™ HP column (GE Healthcare, 17524802). After washing with buffer containing 200 mM Tris, pH 8.0, 500 mM NaCl, 40 mM imidazole, and 5% glycerol, the target protein was eluted using the wash buffer supplemented with 500 mM imidazole. The proteins were concentrated using a 3-kDa centrifugal filter (Merck Millipore, UFC8003) and further purified by using size-exclusion chromatography (Superdex™ 75 Increase 10/300 GL, GE Healthcare, 29148722) in the buffer containing 25 mM HEPES, pH 7.4, 150 mM NaCl, and 0.25% CHAPSO (Sigma, C3649). Protein concentration was determined by using a NanoDrop 2000 spectrophotometer.

## Preparation of 4-HNE-modified C99, C99$^{Lys53Arg}$, and Aβ40 polypeptides

*For C99 and C99$^{Lys53Arg}$*: Aliquots of recombinant C99, or recombinant C99$^{Lys53Arg}$ solution was incubated with a 50-fold excess of (*R*)−4-HNE (mol/mol) in buffer containing 25 mM HEPES, 150 mM NaCl, pH 7.4. The reaction was carried out at 37 °C for 1 h. Excess-free 4-HNE was then extracted three times with 400 µL hexane.

*For Aβ40*: Synthetic Aβ40 (0.004 µmol) was incubated with a five- or ten-fold molar excess of (*R*)−4-HNE in PBS. The reaction was carried out at 37 °C for 1 or 12 h. Excess-free 4-HNE was then extracted three times with 400 µL hexane. Verification was performed using Coomassie Brilliant Blue staining, WB with 4-HNE or C1/6.1 antibodies.

## Extraction and identification of γ-secretase

The extraction and identification of γ-secretase were performed as described by Yueming Li et al. [66]. Briefly, γ-secretase complexes were

isolated from HEK293T cells that were pretreated without or with 10 μM (R)−4-HNE for 4 h. The cells were harvested by centrifugation and lysed in a buffer containing 25 mM HEPES (pH 7.4), 150 mM NaCl, and a protease inhibitor cocktail (Roche). After sonication, cellular debris was removed by centrifugation at $1000 \times g$ for 30 min, then the membrane fractions were collected by ultracentrifugation at $150,000 \times g$. The resulting pellet was then resuspended in the same buffer supplemented with 1% CHAPSO and incubated for 2 h at 4 °C. The suspension was subsequently recentrifuged at $150,000 \times g$ for 1 h, and the protein concentration was determined using the BCA assay. Finally, γ-secretase complexes were identified by WB using antibodies against PS1, PEN2, and APH1.

### In vitro γ-secretase cleavage assay

For the γ-secretase cleavage assay shown in Fig. 4a, WT substrate and 4-HNE-adducted substrate were obtained using the NE-PER Nuclear and Cytoplasmic Extraction Reagents Kit, from N2a-APPswe cells pretreated without or with 10 μM (R)−4-HNE for 4 h. After discarding nuclei, the remaining proteins were used as substrate for the γ-secretase cleavage assay.

For the γ-secretase substrates used in the assays shown in Fig. 4c, recombinant C99 and C99$^{Lys53Arg}$ were prepared as described above.

In the γ-secretase cleavage assay, 30 nM purified γ-secretase (WT or 4-HNE-adducted) was incubated with 12.5 mM substrate (WT or 4-HNE-adducted, substrate from N2a-APPswe cells, C99, or C99$^{Lys53Arg}$) in reaction buffer that contained 25 mM HEPES, pH 7.4, 0.2% CHAPSO, 150 mM NaCl, 0.1% phosphatidylcholine, 0.025% phosphatidylethanolamine, and 0.00625% (w/v) cholesterol. The cleavage reaction was conducted at 37 °C for 12 h. Aβ40 and Aβ42 production were detected using the Human Amyloid β Peptide 1−40 (Aβ1−40) ELISA Kit and Human Amyloid β Peptide 1−42 (Aβ1−42) ELISA Kit, respectively.

### MALDI-TOF MS

Low-salt solutions of Ac-Leu−Lys−Lys−Lys−Gln (0.029 μmol) and (R)−4-HNE (0.132 μmol) were incubated in PBS (0.01 M, pH 7.2) for 1 h at 37 °C. Matrix-assisted laser desorption/ionization time-of-flight mass spectrometry (MALDI-TOF MS) was performed on the samples by the 4800 Plus MALDI TOF/TOF™ Analyzer (AB SCIEX, USA).

### Organelle collection and identification

HEK293T cells were cultured and seeded at a density of $1.6 \times 10^5$ cells per 10-cm dish and subsequently exposed to 2 μM (R)−4-HNE for 24 h. Organelle samples were obtained from these cells, with $3.0 \times 10^7$ cells collected for each compartment, namely the Golgi apparatus and early endosomes. The Minute Golgi Apparatus Enrichment Kit (Invent, GO-037) and Minute Endosome Isolation and Cell Fractionation Kit (Invent, ED-028) were used in accordance with the manufacturer's instructions for the purpose of Golgi and early endosome enrichment, respectively. Following organelle extraction, the resulting pellets containing early endosomes and Golgi were immediately frozen in dry ice and stored at −80 °C for subsequent BCA protein quantification and ELISA.

### Preparation of oligomeric Aβ40 (oAβ40) and fibrillar Aβ40 (fAβ40)

Aβ40 peptides (GL Biochem) were solubilized in 1,1,1,3,3,3-hexafluoro-2-propanol (HFIP; Sigma, 1105228) at 1 mM. The HFIP was evaporated in a fume hood, and the dried peptides were stored at −80 °C. Monomeric Aβ (mAβ) was obtained by resuspending Aβ40 peptides in dimethyl sulfoxide (Sigma) at 1 mM. oAβ40 and fAβ40 were generated by diluting mAβ to 10 mM and 100 mM in DMEM, respectively. The resulting solutions were then incubated overnight at 4 and 37 °C, respectively.

### RNA isolation, reverse transcription, and RT-qPCR

RNA was extracted from cell pellets using TRIzol reagent (Invitrogen, 15596018) according to the manufacturer's instructions. The extracted RNAs were then dissolved in diethyl pyrocarbonate-treated water (Thermo Fisher Scientific, 750023), and their purity and concentration were assessed using a NanoDrop 2000 spectrophotometer. A total of 1000 ng of RNA was reverse transcribed into cDNA using the Prime-Script™ RT Master Mix Kit (Takara, RR036A) following the manufacturer's protocol. For RT-qPCR, triplicates of cDNA samples were loaded for each gene in a total volume of 20 μL, using SYBR Green PCR Master Mix (Takara, RR820A) per the manufacturer's instructions. The triplicate Ct values were averaged, and the data were normalized to the geometric mean for two housekeeping genes [*ACTB* (encoding β-actin) and *GAPDH*] using the Ct delta method ($2^{-\Delta\Delta Ct}$). The primers (Sangon Biotech, Shanghai, China) were used at a concentration of 10 nM, and their sequences are shown in Supplementary Table 9.

### Phagocytosis assays

Microglial phagocytosis was detected using immunofluorescence and flow cytometry assays.

For immunofluorescence, BV2 cells ($5 \times 10^4$ cells/well) were seeded in 24-well plates coated with poly-L-lysine and cultured overnight for adherence; 60 μM of daidzin was added for 48 h. Then, FITC-conjugated latex beads (0.01%, v/v; Sigma, L1030) were preincubated in DMEM containing 0.05% (v/v) FBS at 37 °C for 1 h, and were added to the cells. After incubation for 4 h, immunofluorescence analysis was conducted.

For flow cytometry, BV2 cells ($1 \times 10^5$ cells/well) were seeded in six-well plates and cultured overnight. After the knockdown of Aldh2 for 48 h, oAβ40 (1 μM) was added for 24 h, followed by the addition of preincubated beads for 4 h. Cell pellets were collected by centrifugation at $500 \times g$ for 5 min. Simultaneously, an appropriate amount of BD Cytofix™ Fixation Buffer (BD Biosciences, 554655) was preheated at 37 °C for 5 min. The pellets were then resuspended in 300 μL Fixation Buffer and 300 μL PBS and incubated for 10 min at 37 °C. Subsequently, centrifugation was performed at $500 \times g$ for 5 min, and the cells were resuspended in 1 mL of PBS for flow cytometry analysis using a CytoFLEX flow cytometer (Beckman Coulter, USA). Data were analyzed and plotted using FlowJo software.

### Apoptosis analysis assays

A total of $1 \times 10^6$ SH-SY5Y cells in a 6-cm dish were treated with 2 μM (±) −4-HNE, 2 μM (S)-4-HNE, or 2 μM (R)-4-HNE for 72 h. Following the treatment, the cells were washed with PBS, harvested, and labeled using the FITC Annexin V Apoptosis Detection Kit with PI (Biolegend, 640914), following the manufacturer's instructions. Staining was detected using a CytoFLEX flow cytometer, and the data were analyzed and plotted using FlowJo software. The initial gating of the whole cell population, excluding the debris, was based on FCS-A/SSC-A. The gated singlets were further classified using Annexin V-FITC and PI signals and represented as dot plots. The percentage of apoptotic cells was determined by combining the percentages of early apoptotic cells (Annexin V$^{high}$, PI$^{low}$) and late apoptotic/necrotic cells (Annexin V$^{high}$, PI$^{high}$), as depicted in the graphs.

### Cell proliferation assay

For cell growth assays, SH-SY5Y or BV2 cells were plated in 96-well plates. After drug treatment for the indicated incubation time, CCK8 solution (MCE, HY-K0301) was added for 2 h. The absorbance was measured at 450 nm.

### Protein structure prediction

We used ZDOCK 3.0.2 software to predict the interaction sites between 4-HNE-modified Val46−Lys53(HNE)−Lys55 truncate or natural Val46−Lys55 truncate of C99 and γ-secretase. Natural Val46−Lys55

truncate of C99 was obtained from PDB structure 6IYC [https://www.rcsb.org/structure/6IYC] was used to manually add 4-HNE modification on Lys53 residue and analyze protein structures and align predicted structures.

## Statistical analysis

Data are presented as mean values ± SD. All box plots include the median line, the box indicates the interquartile range, and whiskers indicate minima and maxima. The protein expression level in immunohistochemistry was measured as the mean integrated optical density (IOD) quantified by Image-Pro Plus 6.0 software. Statistical analyses were conducted using SPSS software, with a two-tailed Student's t-test for two groups and a one-way ANOVA for multiple groups with the LSD post-hoc test. Spearman correlation coefficients and ordinal logistic regression were calculated using SPSS software. $P < 0.05$ was considered to indicate statistical significance.

## Reporting summary

Further information on research design is available in the Nature Portfolio Reporting Summary linked to this article.

## Data availability

The mass spectrometry proteomics data generated in this study have been deposited to the ProteomeXchange Consortium via the iProX partner repository[67,68] with the dataset identifier PXD047209 and PXD047210. The raw sequence data reported in this paper have been deposited in the Genome Sequence Archive (Genomics, Proteomics & Bioinformatics 2021) at the National Genomics Data Center (Nucleic Acids Res 2022), China National Center for Bioinformation/Beijing Institute of Genomics, Chinese Academy of Sciences, the accession number HRA006714[69,70], under controlled access. Data are available from the corresponding author upon request and with no delay for data sharing. The published structure of free C99 (PDB: 2LLM) and the structure of C99 bound to γ-secretase (PDB: 6IYC) were used in Fig. 4e. Source data are available on Figshare (https://doi.org/10.6084/m9.figshare.24647757) and provided with the article. Source data are provided with this paper.

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

## Acknowledgements

This study was supported by the National Natural Science Foundation of China (NSFC 81971023 to W.G., 82171332 to X.W., 92353302 to X.W.), STI2030-Major Project (2021ZD0201100 to C.M.), and the CAMS Innovation Fund for Medical Sciences (CIFMS 2021-I2M-1-025 to C.M. and W.G.). The authors thank H.L., M.P. and C.L. for their help in the manuscript preparation. The authors thank the National Human Brain Bank for Development and Function for providing the human postmortem brain tissues, brain donors, and their families for tissue samples used in this study.

## Author contributions

Xia W. and J.W. conducted all the experiments and data analysis, managed all figures, and drafted the manuscript. Y.C. helped perform animal and cell experiments. X.Q. and Xue W. helped prepare postmortem brain tissues and assess AD pathology. S.L. conducted protein–protein docking. W.G. and C.M. conceived, designed, and supervised the study and edited the manuscript.

## Competing interests

The authors declare no competing interests.
