## [Peer Review File NEW · Nature Communications]

The aldehyde dehydrogenase 2 rs671 variant enhances amyloid beta pathologyReviewer #1 (Remarks to the Author):

The manuscript of Wang et al. describes exciting new data on the role of ALDH2 variant (rs671), common in the East Asian population, in Alzheimer's disease. The study does not only explain the potentially conflicting data on the association of this variant with the disease in humans but also provides molecular and cellular basis for how ALDH2 reduced function causes it. This comprehensive study relies on a relatively large collection of brain autopsy samples, multiple cell culture models, two mouse models, and several pharmacological tools.

In short, the study demonstrated that diminished clearance of the aldehyde 4HNE due to reduced ALDH2 activity increases the modification of the A-beta precursor protein, thus altering gamma secretase activity which results in the accumulation of oligomerized A beta 40. The study shows that the modified Abeta aggregates accumulate at the Golgi and that in microglia, this causes increased proinflammatory effect. An exciting analysis of the microglia behavior (using cultured BV2 and histochemistry of human and animal models of AD) indicates that decreased ALDH2 activity impairs microglial Abeta phagocytosis, leading to increased diffusion of toxic Abeta throughout the brain. Finally, the authors identified the differential effect of the racemic 4HNE and the potential molecular explanation for its effect on A beta40 accumulation. All in all, the study covers many aspects – perhaps even too many. Nevertheless, it is easy to follow and written clearly, and the data support their conclusions.

A few comments:

Data analysis:

It is not clear how quantitation of the histochemistry data was conducted. Also, how was potential bias in the analysis prevented (e.g. was the analysis conducted by a reviewer blinded to the experimental conditions?). Also, the immunofluorescence data show a single cell. A lower magnification containing more cells should be provided in the supplementary material and the number (%) of cells showing the phenotype should be indicated.

Writing:

In the summary, introduction, and results, the authors describe the opposing functions of microglia as a double edge sword, providing explanation to the meaning of that only in the discussion. It would be better to refer to these findings as opposing effects on microglial functions and use the expression 'double edge sword' once in the discussion.

Abstract – 'clearly' should be removed from the last sentence.

Results – The study did not 'enroll' the participants to the study. Rather, the author used existing autopsy samples from the Brain Bank. The language should be corrected accordingly.

In several places, the authors use 'a decrease of ALDH2 levels', the 'decreased ALDH2 levels', etc. It should be 'lower ALDH2 protein levels'

On p. 8, upper paragraph, the authors discuss ADH1B activity. However, only levels of the enzyme were assessed. Also, Daidzein inhibits (not suppressed) ALDH2 enzyme activity.

In the legend of Fig. 6, state what siNC; it is explained only in Fig. 7.

Fig. 1: indicates that 6E10 is an anti A beta plaque antibody.

P.22 'both disrupted the opposing effects' should be 'disrupted both opposing effects'.

Discussion of Alda-1 on Page 24: The authors speculate why the clinical trial by ALDEA was terminated based on an unreviewed comment on the web. In fact, ALDEA used another alda (not Alda-1) in a small phase 1 clinical study. The study was completed and not terminated for toxicity. Rather, the work was terminated because the inventors realized that the clinical trial size needed to be bigger than they initially thought. They pulled out the rest of the investment and folded ALDEA. Since then, the IP was licensed to Foresee, and a clinical trial for Alda in a pediatric

indication, Fanconi Anemia, is ongoing (NCT04522375) after a successful safety study was completed.

The use of Alda-1 in this report: How Alda-1 or daidzein was dosed in the mice is not provided in the Method section. Perhaps the noted toxicity was due to a single intraperitoneal injection of a very high drug dose or the vehicle used (also not indicated in the manuscript).

In published studies, Alda-1 was dosed in WT mice for several months and was found to be safe, for example, in models of post-myocardial infarction heart failure, in a model of Parkinson's disease, and in a chronic model of ethanol-induced neurotoxicity. In all these studies, Alda-1 was delivered at 10mg/kd/day using a slow delivery via a subcutaneous Alzet pump.

Relevant to the current report, examining the effect of sustained treatment with Alda-1 in ALDH2*2 knock-in mice, Alda-1, dissolved in vehicle (50% PEG-400, 50% DMSO), was delivered using 4-week osmotic pumps at 10 mg/kg/day (0.11 μ l daily volume). Alda-1 was delivered for 12 weeks with no adverse effects. Instead, Alda-1 sustained treatment using the above method abrogated neuroinflammation, reduced impaired mitochondrial functions, reduced 4HNE and malondialdehyde accumulation, decreased tau phosphorylation, Abeta levels, and synaptic loss. Alda-1 also decreased caspase activation and suppressed ethanol-induced neuroinflammation in both WT and ALDH2*2 knock-in mice, as measured by reduced production of TNFa, IL6, MCP-1, IL1a, C1q, and IL1b (See Joshi... Mochly-Rosen; *Acta Neuropathol Commun.* 2019 Dec 12;7(1):190. doi: 10.1186/s40478-019-0839-7.) This aforementioned study is relevant to the current manuscript and should be discussed as well.

Reviewer #2 (Remarks to the Author):

The manuscript titled "Aldehyde dehydrogenase 2 rs671 variant enhances Alzheimer's disease pathology" by Wang et al describes a comprehensive genetic, in vitro and in vivo study to link the rs671 A variant to Alzheimer's disease (AD), but not as a genetic risk factor, but instead as a modifier variant resulting in increased amyloid beta pathology identified in post-mortem brain tissue. A major strength of this study is the large East Asian ancestry cohort used to directly compare genotype with neuropathology.

There have been multiple recent studies, particularly in East Asian populations, investigating the link between rs671 genotype and AD, including meta-analyses of these studies (example: doi.org/10.1186/s13643-022-02050-y). Wang et al have not sought to replicate these studies, but instead investigating the potential biological relevance of this genetic variant in relation to AD. In vitro and in vivo studies determine a functional reduction/inhibition of ALDH2 enzyme (in A variant carriers – GA/AA genotypes) resulting in impaired microglia action leading to aggregation through the increased spreading of AB plaques. The paper is well-written, and the data is mostly well presented.

Major comments:

1. A large number of studies have been conducted investigating the ALDH2 rs671 variant in multiple diseases such as listed in the following publication: doi.org/10.1186/s13643-022-02050-y. It would be important for the authors to mention that how the rs671 variant has been identified as a risk factor and/or susceptibility locus in multiple diseases, not just in relation to AD.

2. With such a large cohort to select from, as a reviewer it is frustrating to see a study design not matched for sex and age, especially since the authors conducted such a comprehensive and time-consuming IHC of 8 regions per patient selected. The GG and GA genotype cohorts selected for IHC both do not have 1:1 F:M ratio selected (the AA genotype cohort is 1:1 F:M matched). I also note that the average age for IHC selected samples is GG: 81.5 years, GA: 83 years and AA: 89.75 years. My concern with their study design is that, in general, A β deposition increases with age, and the IHC and ELISA cohorts chosen have increasing age showing increasing A β pathology.

Can the authors please include an additional table or information to demonstrate that the samples selected were representative of the genotype cohort. For example, what is the average age/stdev of AA genotype AD patients compared to the 4 selected for IHC and ELISA. Can the authors please also clarify why the cohorts were not matched for sex or age.

3. Since the authors have made a point in their study design not to include heavy drinkers, it would be nice to see a brief discussion point (maybe 2 sentences) about the role of ALDH2 rs671 in alcohol metabolism and AD, since this is a key published function of the rs671 variant, particularly in East Asian populations.

4. As mentioned above, a strength to this study is using pathology-diagnosed AD (vs clinician-diagnosed AD). If the data is available, co-pathologies would be interesting to include in Supplementary Table 1 and whether there is any correlation with rs671 genotype.

5. There is a contradictory statement on page 25 in the Discussion "However, there are not enough AA individuals to demonstrated the association with AD pathology change", yet in multiple other locations throughout the text including in the title of the manuscript, the authors describe that the "ALDH2 rs671 variant positively correlates with A β pathology".

6. Do East Asian populations generally have elevated AB pathology compared to non-East Asian populations? If so, would the increased prevalence of the A allele account for this?

7. Given the increasing genotype-phenotype correlations that are being identified through large biobank efforts, this is a topical analysis. Can the authors please include a discussion point on what is the clinical relevance of this genotype-phenotype correlation. Could it be useful for subgrouping for clinical trials or post-hoc analysis of clinical trial efficacy?

8. There is no Table legend for supplementary figure 1 (4-page pdf) provided with manuscript submission. What is the column "with other CNS disorders"? What other CNS disorders were investigated during post-mortem pathology.

9. It is not clear to me the value of including RNA-sequencing data since there are only 2/50 included participants with the AA genotype, and none overlap the samples included in this study. Can additional AA genotype patients undergo RNA-seq?

10. Can the author please comment on why "the effect of ALDH2 rs671 polymorphism should be properly adjusted when ranking the A β pathology score on postmortem brains". This was not clear to me.

Minor comments:

11. Please include HUGO nomenclature for any genetic variant at first mention in the manuscript (nucleotide and protein change) and glossary, including accession number.

12. Please consider using the word "sex" instead of "gender" throughout the text/figures/tables if you are referring to the patient's sex chromosome genotype (i.e., XX or XY).

13. Repetitive text at end of page 2 in Glossary "Neuropathological evaluation" section.

14. Please include additional references throughout glossary where appropriate when presenting previously published work. Specific examples include the genotype-enzymatic activity on page 1, Cognitive functional assessment on page 4, A β 40 and A β 42 generation on page 5,

15. Supplementary File Table i. please remove the word "change" in table and include accession number.

16. Extended Data Fig 1. Please include a detailed description of abbreviations where appropriate in the Figure legend.

17. Supplementary Table 4. What is NC?

18. A small formatting comment for supplementary table that goes across 4 pages (Supplementary table 1?), please put headings on each page and table legend on pdf.

Reviewer #3 (Remarks to the Author):

Review of "Aldehyde dehydrogenase 2 rs671 variant enhances Alzheimer's disease pathology" by Wang et al.

The authors have in a screen of 329 human brain samples of investigated the role of ALDH2 rs671 polymorphism in AD. This genetic variant has previously been shown to present a risk for hypertension, diabetes and coronary heart disease in the Asian population. For AD, the impact of this polymorphism have so far been contradictory.

The investigation was initiated on a population biobank consisting of 59% male and 41% female samples, on which genotyping was conducted. In turn, amyloid plaque assessment was done to find potential correlation between alleles and phenotype. Further, detailed analysis on Ab levels was done with ELISA. Next, the authors investigated the effects of the Aldh2-gene on Ab-peptide ratios in the APPSwe mouse model of AD. This was done both with a knock-out and by pharmacological modulation of the Aldh2. The author then investigate the potential accumulation of 4-HNE on Ab-peptide ratios. Adduction of 4-HNE to Ab-peptides are found with mass spectrometry, and phenotypic effects were observed with an in vivo assay using HEK293 cells. Overall the manuscript present many interesting findings and connect a large screening study with detailed analysis of molecular mechanisms that highlight the importance of Aldh2 in AD.

However, I have major concerns regarding the proteomic data that the authors must address.

I am confused by how the authors present their proteomic data with volcano plots in extended data figs 6 and 14. It is not clear from the figures how many replicates were measured with proteomics. There are apparently no statistics involved in determining whether a protein has been up- or downregulated. Is extended data figure 6 based on only a single replicate of each condition? Rather, the authors choose an arbitrary cutoff at $\text{abs}(\text{foldchange}) > 1.25$ in extended data figure 6 and $\text{abs}(\text{foldchange}) > 1.17$ in extended data figure 14. This is not an acceptable method to show significant effects in differential expression, because a protein with only $\text{logFC} = 0.5$ can have $\text{adj.p.value} > 0.0001$, but something $\text{logFC} = 3$ can have $\text{adj.p.value} = 0.5$. Instead, the authors must reprocess the data with proper statistical methods, such as Student's t-test or suitable moderated t-tests using either the R-packages limma or DESeq2. The appropriate statistical analysis must be done, including multiple testing correction such as the Benjamini-Hochberg procedure, where volcano plots should show logFC on the x-axis and $\text{log}_{10}(-\text{adj.p.value})$ on the y-axis.

Data availability: The proteomic data must be submitted to PRIDE (public repository). This has not been done.

Response Letter

REVIEWER COMMENTS

Reviewer #1 (Remarks to the Author):

The manuscript of Wang et al. describes exciting new data on the role of ALDH2 variant (rs671), common in the East Asian population, in Alzheimer's disease. The study does not only explain the potentially conflicting data on the association of this variant with the disease in humans but also provides molecular and cellular basis for how ALDH2 reduced function causes it. This comprehensive study relies on a relatively large collection of brain autopsy samples, multiple cell culture models, two mouse models, and several pharmacological tools.

In short, the study demonstrated that diminished clearance of the aldehyde 4HNE due to reduced ALDH2 activity increases the modification of the A-beta precursor protein, thus altering gamma secretase activity which results in the accumulation of oligomerized A beta 40. The study shows that the modified Abeta aggregates accumulate at the Golgi and that in microglia, this causes increased proinflammatory effect. An exciting analysis of the microglia behavior (using cultured BV2 and histochemistry of human and animal models of AD) indicates that decreased ALDH2 activity impairs microglial Abeta phagocytosis, leading to increased diffusion of toxic Abeta throughout the brain. Finally, the authors identified the differential effect of the racemic 4HNE and the potential molecular explanation for its effect on A beta40 accumulation. All in all, the study covers many aspects – perhaps even too many. Nevertheless, it is easy to follow and written clearly, and the data support their conclusions.

A few comments:

Data analysis:

It is not clear how quantitation of the histochemistry data was conducted. Also, how was potential bias in the analysis prevented (e.g. was the analysis conducted by a reviewer blinded to the experimental conditions?).

Response: In this study, immunohistochemistry (IHC) staining included two parts: A β plaque staining with anti-6E10 antibody in human brains (Fig. 1c, Supplementary Fig. 2a) and APP/PS1 mice brains (Fig 2e, Supplementary Fig. 14a), 4-HNE staining with anti-4-HNE antibody in human brains and ALDH2^{-/-} mice (Fig. 3b-c).

For A β plaque staining, percent of amyloid plaque area was quantified with 3 randomly selected fields in each slide.

For 4-HNE staining, average IOD with at least 3 randomly selected fields was calculated in each slide.

The quantification of IHC staining was conducted in a blinded fashion by the author (Y.C.) who was blinded to the experimental groups using Image Pro Plus software.

We added the information in the Methods ‘Tissue processing for immunohistochemistry’ section.

Also, the immunofluorescence data show a single cell. A lower magnification containing more cells should be provided in the supplementary material and the number (%) of cells showing the phenotype should be indicated.

Response: To verify the results in Fig. 6a, we reperformed the immunofluorescent staining on WT 293T and (R)-4-HNE treated 293T at a final concentration of 2 μ M (R)-4-HNE for 24h. After double staining with anti-C1/6.1 (staining for APP) and anti-RCAS1 (staining for Golgi), lower magnification images containing more cells were added in the Supplementary Fig. 11a. The statistical analysis of APP⁺RCAS1⁺ cells were indicated in this figure.

To verify the results in Fig 6i, we first prepared VPS35 knock down and control 293T cells using siVPS35 and relative siNC. Then the negative control (siNC-193T) and VPS35 knock down (siVPS35-293T) cells were treated with PBS or 2 μ M of (R)-4-HNE for 24h. Double staining for APP (anti-C1/6.1) and Golgi (anti-RCAS1) were conducted. Lower magnification images and statistical analysis of APP⁺RCAS1⁺ cells were provided in the Supplementary Fig. 11f.

The revised Supplementary Fig. 11a,f were as following:

Supplementary Fig. 11a. A lower magnification of Fig. 6a. Representative confocal images of HEK293T cells stained with anti-C1/6.1 (red, APP) and anti-RCAS1 (green, Golgi apparatus). HEK293T cells were pretreated with PBS or 2 μ M of (R)-4-HNE for 24 h. Scale bar, 30 μ m.

Supplementary Fig. 11f. A lower magnification of Fig. 6i. VPS35 levels were suppressed by RNA silencing in HEK293T cells, and cells were then treated with PBS or 2 μ M (R)-4-HNE for 24 h. Representative images of double staining with anti-C1/6.1 (red, APP) and anti-RCAS1 (green, Golgi apparatus) antibodies in the above cells. Scale bar, 30 μ m.

Writing:

In the summary, introduction, and results, the authors describe the opposing functions of microglia as a double edge sword, providing explanation to the meaning of that only in the discussion. It would be better to refer to these findings as opposing effects on microglial functions and use the expression ‘double edge sword’ once in the discussion.

Response: Thanks for your kind suggestion. We now provide more clear expression using opposing effects of microglia and replaced the expression “double edge sword” in the Introduction and Results sections.

Abstract – ‘clearly’ should be removed from the last sentence.

Response: We removed “clearly” from the last sentence in the Abstract. The sentence was corrected to “We thus defined the relationship between *ALDH2* rs671 polymorphism and AD, and found *ALDH2* rs671 as a key regulator of A β 40 or A β 42 generation”.

Results – The study did not ‘enroll’ the participants to the study. Rather, the author used existing autopsy samples from the Brain Bank. The language should be corrected accordingly.

Response: In the Results section, we removed the ‘enrolled’. The sentence was now revised to ‘...a total of 329 participants from the Human Brain Bank were included for *ALDH2* rs671 sequencing,...’.

In several places, the authors use ‘a decrease of *ALDH2* levels’, the ‘decreased *ALDH2* levels”, etc. It should be ‘lower *ALDH2* protein levels’

Response: To provide a clear description, we revised the ‘a decrease of *ALDH2* levels’, ‘a decrease of *ALDH2* activities, and ‘the decreased *ALDH2* activities’ to be ‘lower *ALDH2* protein levels’, or ‘lower *ALDH2* activities’ in several places throughout the whole manuscript.

On p. 8, upper paragraph, the authors discuss *ADH1B* activity. However, only levels of the enzyme were assessed. Also, Daidzein inhibits (not suppressed) *ALDH2* enzyme activity.

Response: *ADH1B* is the first enzyme in alcohol metabolism, the upstream gene of *ALDH2*, that catalyzes alcohols into aldehydes.

ALDH2 is a key mitochondrial aldehyde dehydrogenase, which catalyze acetaldehyde to acetic acid. *ALDH2* rs671 G>A (Glu487Lys) leads to a substantial decrease in dehydrogenase activity. According to previous reports, we used 2 methods to reduce the enzyme activity of *ALDH2* in SH-SY5Y and N2a-APPsw cells: RNA silencing with si*ALDH2*, or dardzin treatment. Daidzin is a commonly used *ALDH2* antagonist which specifically inhibited *ALDH2* enzyme activity. We then detected/verified the lower/reduced *ALDH2* enzymatic activity using the colorimetric Mitochondrial Aldehyde Dehydrogenase

(ALDH2) Activity Assay Kit (Abcam, ab115348, Cambridge, MA). And we verified decrease of ALDH2 activity in SH-SY5Y and N2a-APPswe cells in both methods. The results were provided in Fig. 2g and Supplementary Fig. 5i,m.

Fig. 2g. ALDH2 knockdown by siRNA or daidzin (60 μ M) treatment for 48 h in SH-SY5Y cells. Enzymatic activity of ALDH2 in cell lysates, measured over 120 min by using a Mitochondrial Aldehyde Dehydrogenase (ALDH2) Activity Assay Kit.

Supplementary Fig. 5i, m. N2a-APPswe cells with Aldh2 knockdown (i) and with daidzin treatment (60 μ M) for 48 h (m). Enzymatic activity of Aldh2 in cell lysates, measured over 120 min.

In the legend of Fig. 6, state what siNC; it is explained only in Fig. 7.

Response: siNC is the negative control small interfering RNA, randomly scrambled sequences that do not target any genes. In this study, the siNC sequence is shown in Methods “Small interfering RNA transfection” section. we used the same sequence as siNC in human RNA silencing and mouse RNA silencing. We added “siNC, negative control small interfering RNA” in Fig. 2 legend, Fig. 6 legend and Fig. 7 legend.

Fig. 1: indicates that 6E10 is an anti A beta plaque antibody.

Response: We provided the indication “6E10 antibody (anti- β -amyloid 1-16 antibody)” in Fig. 1b legend.

P.22 'both disrupted the opposing effects' should be 'disrupted both opposing effects'.

Response: We revised the sentence to “This study demonstrated that reduced ALDH2 activity disrupted both opposing effects of microglia” in the last paragraph in P22.

Discussion of Alda-1 on Page 24: The authors speculate why the clinical trial by ALDEA was terminated based on an unreviewed comment on the web. In fact, ALDEA used another alda (not Alda-1) in a small phase 1 clinical study. The study was completed and not terminated for toxicity. Rather, the work was terminated because the inventors realized that the clinical trial size needed to be bigger than they initially thought. They pulled out the rest of the investment and folded ALDEA. Since then, the IP was licensed to Foresee, and a clinical trial for Alda in a pediatric indication, Fanconi Anemia, is ongoing (NCT04522375) after a successful safety study was completed.

The use of Alda-1 in this report: How Alda-1 or daidzein was dosed in the mice is not provided in the Method section. Perhaps the noted toxicity was due to a single intraperitoneal injection of a very high drug dose or the vehicle used (also not indicated in the manuscript).

In published studies, Alda-1 was dosed in WT mice for several months and was found to be safe, for example, in models of post-myocardial infarction heart failure, in a model of Parkinson's disease, and in a chronic model of ethanol-induced neurotoxicity. In all these studies, Alda-1 was delivered at 10mg/kd/day using a slow delivery via a subcutaneous Alzet pump.

Relevant to the current report, examining the effect of sustained treatment with Alda-1 in ALDH2*2 knock-in mice, Alda-1, dissolved in vehicle (50% PEG-400, 50% DMSO), was delivered using 4-week osmotic pumps at 10 mg/kg/day (0.11 µl daily volume). Alda-1 was delivered for 12 weeks with no adverse effects. Instead, Alda-1 sustained treatment using the above method abrogated neuroinflammation, reduced impaired mitochondrial functions, reduced 4HNE and malondialdehyde accumulation, decreased tau phosphorylation, Abeta levels, and synaptic loss. Alda-1 also decreased caspase activation and suppressed ethanol-induced neuroinflammation in both WT and ALDH2*2 knock-in mice, as measured by reduced production of TNFa, IL6, MCP-1, IL1a, C1q, and IL1b (See Joshi... Mochly-Rosen; Acta Neuropathol Commun. 2019 Dec 12;7(1):190. doi: 10.1186/s40478-019-0839-7.) This aforementioned study is relevant to the current manuscript and should be discussed as well.

Response: Thanks for your detail introduction and suggestion.

In this study, the dose of Alda-1 used APP/PS1 mice is 15 mg/kg/day of body weight, via intragastric infusion (*iG*) for two months. In the Methods “Animal experimental models and drug treatment” section, we added the missing details and clearly stated the administration of Alda-1 and daidzin in mice, including the reagents used for dissolution, the dosing and procedures of administration. This added description was as shown below:

Drug treatment and brain tissue preparation. Alda-1 was dissolved in 50% DMSO/50% PEG-400 (v/v). Daidzin was dissolved in 10% DMSO/40% PEG-400/ 5% Tween-80/45% saline (v/v/v/v). Four 2-month-old male APP/PS1 mice were administered physiological saline via intragastric infusion (*iG*) as the control group, while four mice were given daidzin (at a dose of 150 mg/kg/day of body weight) or Alda-1 (at a dose of 15 mg/kg/day of body weight) via *iG* for 2 months.

The dissolving strategy used for Alda-1 and Daidzin in this study was according to the manufacture instructions. We noted that our dissolving vehicle for Alda-1 is consistent to the previous report.

In proteomics analysis part, we reorganized the screening of differentially expressed proteins induced by Daidzin or Alda-1 in mouse neuron N2a-APP_{swe} and in mouse microglia BV2 cells. According to previous published papers, we set adjusted $P < 0.05$ and protein expression fold change < 0.83 or > 1.20 as filtering criteria. We found that daidzin treatment mainly affected fatty acid metabolism in N2a-APP_{swe} (Fig. 3a). Alda-1 did not significantly alter proteins profiling in N2a-APP_{swe} (Supplementary Fig. 14c). In BV2, Alda-1 treatment altered expression of proteins related to metabolisms of lipid or steroids (Supplementary Fig. 14f-h). After bioinformatic analysis, we did not find significant alteration of apoptosis related proteins. So, we revised the description of Alda-1 in Discussion section.

In this section, we added the discussion of Alda-1 in APP/PS1 mice reported by Joshi et al. The added and revised discussion are as follows:

Several small molecule activators have been reported to improve the activity of mutated ALDH2, represented by Alda-1. Yang et al. found that Aldh2 overexpression improved cognitive function of APP/PS1 mice⁵⁶. *ALDH2* rs671 G>A greatly increased aldehydic load and exacerbated ethanol-induced neuropathology change. In *ALDH2* AA-homozygous knock-in mice, Alda-1 administration significantly blunted the ethanol-induced increases in A β and neuroinflammation *in vivo* and in primary neurons and astrocytes, with underlying mechanism of effective clearance of ethanol-derived acetaldehyde by Alda-

1⁴¹. Without ethanol consumption, we found Alda-1 administration also slightly reduced the plaque deposition area and mildly decreased the levels of A β 40 and A β 42 in APP/PS1 mice, though with no significance (Supplementary Fig. 14a–b). *In vitro*, Alda-1 did not significantly affect the wide-type neuron with no significant altering of protein expression profiling in N2a-APP_{swe} (Supplementary Fig. 14c). Wide-type microglia BV2 showed inhibited proliferation (Supplementary Fig. 14d-e) but unaffected phagocytic ability (Supplementary Fig. 12h) after treatment of 20 μ M or more Alda-1. The altered metabolisms of lipid or steroids (Supplementary Fig. 14f-h, Supplementary Table 8) may account for the inhibition effect induced by Alda-1.

Reviewer #2 (Remarks to the Author):

The manuscript titled “Aldehyde dehydrogenase 2 rs671 variant enhances Alzheimer’s disease pathology” by Wang et al describes a comprehensive genetic, in vitro and in vivo study to link the rs671 A variant to Alzheimer’s disease (AD), but not as a genetic risk factor, but instead as a modifier variant resulting in increased amyloid beta pathology identified in post-mortem brain tissue. A major strength of this study is the large East Asian ancestry cohort used to directly compare genotype with neuropathology.

There have been multiple recent studies, particularly in East Asian populations, investigating the link between rs671 genotype and AD, including meta-analyses of these studies (example: doi.org/10.1186/s13643-022-02050-y). Wang et al have not sought to replicate these studies, but instead investigating the potential biological relevance of this genetic variant in relation to AD. In vitro and in vivo studies determine a functional reduction/inhibition of ALDH2 enzyme (in A variant carriers – GA/AA genotypes) resulting in impaired microglia action leading to aggregation through the increased spreading of AB plaques. The paper is well-written, and the data is mostly well presented.

Major comments:

1. A large number of studies have been conducted investigating the ALDH2 rs671 variant in multiple diseases such as listed in the following publication: doi.org/10.1186/s13643-022-02050-y. It would be important for the authors to mention that how the rs671 variant has been identified as a risk factor and/or susceptibility locus in multiple diseases, not just in relation to AD.

Response: Thank you for these critical comments. Indeed, rs671 mutation was linked to multiple diseases, suggesting it leads to a significant change of ALDH2 and the importance of ALDH2 in many disease related pathways. We added the association between ALDH2 rs671 polymorphism and multiple diseases in the Introduction section. The added sentence is as follows:

A large number of studies demonstrated increased association between *ALDH2* rs671 polymorphism and many diseases^{4,5}, including elevated risk of cancers⁶ and cerebral vascular disease⁷ after alcohol consumption, opposite effects in varied cardiovascular diseases³, in *ALDH2* rs671 A-allele carriers.

2. With such a large cohort to select from, as a reviewer it is frustrating to see a study design not matched for sex and age, especially since the authors conducted such a comprehensive and time-consuming IHC of 8 regions per patient selected. The GG and GA genotype cohorts selected for IHC both do not have 1:1 F:M ratio selected (the AA genotype cohort is 1:1 F:M matched). I also note that the average age for IHC selected samples is GG: 81.5 years, GA: 83 years and AA: 89.75 years. My concern with their study design is that, in general, A β deposition increases with age, and the IHC and ELISA cohorts chosen have increasing age showing increasing A β pathology.

Can the authors please include an additional table or information to demonstrate that the samples selected were representative of the genotype cohort. For example, what is the average age/stdev of AA genotype AD patients compared to the 4 selected for IHC and ELISA. Can the authors please also clarify why the cohorts were not matched for sex or age.

Response: To exclude the influence of age on A β pathology, we added 4AA individuals, 14 GA individuals, and 14 GG individuals for IHC staining of A β plaques and Elisa of A β 40 and A β 42 contents. In this assay, we selected AD individuals in each genotype for IHC A β staining and Elisa. Thus, we can exclude the very less or no positive A β staining in those with A score of 0. For AA genotype, due to the very low prevalence (3%), our cohort includes only 14 AA individuals, with 8 were pathology-AD individuals. So, we only added 4 additional AA individuals for IHC and Elisa assays. Totally, 8 AA individuals, 18 GA, and 18 GG were used for IHC (in 8 brain regions) and Elisa (in 3 brain regions) statistics. Now the sex/age are matched in the three groups for IHC and Elisa (shown in the below Table R1.1).

We also calculated the average age and sex distribution in GG, GA, AA genotype groups of the total 469 individuals included in the study. The results were shown in the Table R1.1 below.

Tabel R1.1 Distribution of sex and age of individuals included in this study.

	total enrolled			IHC/Elisa		
rs671	AA	GA	GG	AA	GA	GG
Number	14	123	332	8	18	18
Sex	5M:9F	79M:44F	183M:149F	3M:5F	9M:9F	9M:9F
Age, y	80.36 ± 10.83	81.04 ± 13.48	77.86 ± 14.57	84.62 ± 9.76	84.78 ± 4.40	82.89 ± 7.76

3. Since the authors have made a point in their study design not to include heavy drinkers, it would be nice to see a brief discussion point (maybe 2 sentences) about the role of ALDH2 rs671 in alcohol metabolism and AD, since this is a key published function of the rs671 variant, particularly in East Asian populations.

Response: Thanks for your suggestion. Alcohol consumption is a risk factor for health. While in rs671 GA/AA individuals, the lower activity of dehydrogenase ALDH2 would drastically slow down the metabolism of alcohol-derived acetaldehyde, which is more toxic for cells with the active aldehyde group in the molecule. So, alcohol consumption aggravates the aldehyde load in rs671 A carriers. We searched several published papers and added their results and discussions that associated with alcohol consumption and AD in the second paragraph of Discussion section. The added discussion is as follows:

Low concentration of ethanol shows protective effects against A β toxicity in hippocampal neurons³⁹ and cardiac-cerebral vascular disease in GG genotype individuals^{7,40}. Excessive ethanol exposure is detrimental to the brains and is a higher risk factor for AD³⁹. *ALDH2* rs671 G>A greatly reduces alcohol metabolism inducing toxic aldehyde load. Joshi et al. demonstrated aggravated neuropathology in *ALDH2* AA-allele brains than GG brains after chronic alcohol consumption in mice⁴¹.

4. As mentioned above, a strength to this study is using pathology-diagnosed AD (vs clinician-diagnosed AD). If the data is available, co-pathologies would be interesting to include in Supplementary Table 1 and whether there is any correlation with rs671 genotype.

Response: We agree with the reviewer's opinion about the correlation of co-pathologies with rs671 polymorphism.

While our study is not a prospective cohort study. We used the human brain tissues which were already preserved in the human brain bank. These donors did not receive cognitive function tests before death. Only Ecog Insider Questionnaire were filled out by their immediate kin[1]. The average Ecog ratings were based on a four-point scale, 0-4. Cognitively normal was defined as ECog \leq 1.0; mild cognitive impairment as ECog 1.0-2.0; and dementia as ECog $>$ 2.0[2].

In our study, we obtained the average Ecog scores of 303 donors and analyzed the correlation between rs671 gene polymorphism and the average Ecog score. The average Ecog score of each individual was provided in Supplementary Table 1. Though the ordinal logistic regression analysis showed no higher risk of rs671 polymorphism on the Ecog score (Fig 1a), we found an increased proportion of individuals with average Ecog score $>$ 2 in GA/AA genotype populations. The results were shown in Table R1.2 below.

Table R1.2. The distribution of average ECog score in 303 postmortem brain donors.

rs671 genotype	n	ECog score			
		1.0-1.9	2.0-2.9	3.0-4.0	percentage of 2.0-4.0 in each genotype (%)
GG	210	173	16	21	17.6
GA	82	64	7	11	22.0
AA	11	9	0	2	18.2
GA/AA	93	73	7	13	21.5
total	303	246	23	34	18.8

Reference:

1. Qiu, W., et al. Standardized Operational Protocol for Human Brain Banking in China. *Neurosci. Bull.* 35(2), 270-276 (2019).
2. Yang, Q., et al. Correlations Between Single Nucleotide Polymorphisms, Cognitive Dysfunction, and Postmortem Brain Pathology in Alzheimer's Disease Among Han Chinese. *Neurosci. Bull.* 35(2), 193-204 (2019).

5. There is a contradictory statement on page 25 in the Discussion “However, there are not enough AA individuals to demonstrated the association with AD pathology change”, yet in multiple other locations throughout the text including in the title of the manuscript, the authors describe that the “ALDH2 rs671 variant positively correlates with A β pathology”.

Response: In our data, we found that G>A mutation leads to exacerbated A β pathology in human brains, thus we say “ALDH2 rs671 variant positively correlates with A β pathology”. Besides, we found in the 14 rs671-AA carriers, more proportions (8/14, 57%) are pathology-AD, this proportion is higher than that in GG carriers (115/332, 35%). In this part, we want to convey that the rs671 AA genotype is potentially linked to higher AD risk. However, based on such a small sample size, drawing a conclusive link between AA and AD risk lacks sufficient persuasive evidence. So, in the page 25, we revised the original sentence to “However, there are not enough AA individuals to establish AA as a risk factor for AD”.

6. Do East Asian populations generally have elevated AB pathology compared to non-East Asian populations? If so, would the increased prevalence of the A allele account for this?

Response: Thank you for your interesting opinion.

We checked published papers and found no related finding, we speculate that the A β pathology is not generally elevated in East Asian populations than non-East Asian populations, To the best of my knowledge, this manuscript is the first study investigating the relationship between the *ALDH2* rs671 polymorphism and A β pathology. In non-East Asian populations, the rs671 A-allele prevalence is low (less than 5%), and there have been no reports of rs671 polymorphism being associated with A β pathology. Although there are no reported evidences about this conclusion, a comparison study between different ethnic populations is interesting and required.

The onset and pathological changes of Alzheimer's disease (AD) are induced by multiple factors, such as mutations in APP, PS1, APOE, et al, each may elevate the risk of AD onset and the severity of A β pathology. It is not sufficient to assess the extent of AD pathology in the entire population solely based on changes only at the *ALDH2* rs671 locus; a comprehensive consideration of multiple risk factors or protective factors is necessary. The findings in this study only apply to individuals with the GA/AA genotypes and cannot be extrapolated to the entire East Asian population.

7. Given the increasing genotype-phenotype correlations that are being identified through large biobank efforts, this is a topical analysis. Can the authors please include a discussion point on what is the clinical relevance of this genotype-phenotype correlation. Could it be useful for subgrouping for clinical trials or post-hoc analysis of clinical trial efficacy?

Response:

Alzheimer's disease (AD) is a systemic condition with various clinical and pathological manifestations. Our study specifically focused on A β pathology. The tau phosphorylation and neuroinflammation are another important pathological feature. In our study, we found rs671 A-allele is a risk factor for A β pathology, but it cannot be asserted that rs671 is a risk factor for the AD phenotypes. For instance, APOE4 is a risk factor for AD, but it does not imply a strong correlation between APOE4 and AD onset, that is, individuals carrying APOE4 may not necessarily develop AD. Therefore, there is no direct genotype-phenotype relationship between the rs671 AA genotype and the AD phenotype.

The rs671 polymorphism has potential to be utilized for subgroups, holding significant implications for clinical trials, especially those targeting the clearance of A β oligomers and plaques. As an increased presence of A β plaques is detrimental for brains. In this study, we found more plaques deposits and elevated A β 40/42 ratio in human brains with GA and AA genotypes compared to GG genotype. Thus, plaques with distinct compositions may result in development of antibodies designed to clear A β plaques or oligomers varying between genotypes. Also, different proportions of A β 40 and A β 42 peptides would exhibit varying aggregation patterns. Consequently, the antibodies used to clear these plaques or oligomers vary, highlighting the need for the development of genotype-specific antibodies tailored to different compositions. We added a conclusive and prospective sentence about this opinion in the Conclusion section.

8. There is no Table legend for supplementary figure 1 (4-page pdf) provided with manuscript submission. What is the column “with other CNS disorders”? What other CNS disorders were investigated during post-mortem pathology.

Response: We added the Supplementary Table 1 legend in each page of the table. We also added the list of other CNS diseases as the footnote in Supplementary Table 1. The excluded other CNS diseases include Parkinson's disease, cerebral amyloid angiopathy, brain tumors, Amyotrophic Lateral Sclerosis, stroke, cerebral hemorrhage, brain tumors, schizophrenia.

9. It is not clear to me the value of including RNA-sequencing data since there are only 2/50 included participants with the AA genotype, and none overlap the samples included in this study. Can additional AA genotype patients undergo RNA-seq?

Response: ALDH2 exerts its functions through the protein. In this study, we mainly focus on protein-level changes.

Previous researchers reported that the G>A mutation does not alter ALDH2 mRNA levels, and affects the ALDH2 protein stability and decreases ALDH2 protein levels. We also found lower ALDH2 protein levels in GA/AA brains by western blot, consistent with reported findings.

The RNA extracted from frozen postmortem human brain samples exhibited generally lower RIN (RNA Integrity Number) values. It is challenging to exclude or mitigate potential biases introduced by low RIN values. We utilized the previously published RNA-seq data from our team's work (Supplementary Fig. 2e) as a supplementary data of our proteomic result. The mRNA levels of ALDH2 showed no change in GG, GA, and AA brains, albeit with a small sample size and low RNA RIN value. This data is consistent with published findings.

10. Can the author please comment on why “the effect of ALDH2 rs671 polymorphism should be properly adjusted when ranking the A β pathology score on postmortem brains”. This was not clear to me.

Response: What we want to convey here is that ALDH2 rs671 A allele plays an important role in AD pathological changes. This is repetitive to the second sentence in the same paragraph. We deleted this sentence here.

Minor comments:

11. Please include HUGO nomenclature for any genetic variant at first mention in the manuscript (nucleotide and protein change) and glossary, including accession number.

Response: According to the Sequence Variant Nomenclature of Human Genome Variation Society (HGVS) and Human Genome Organisation (HUGO), we added the formal name of genes and the standard

description of rs671 and rs1229984, including accession number, nucleotide and amino acid changes, in the manuscript and the Glossary file.

12. Please consider using the word “sex” instead of “gender” throughout the text/figures/tables if you are referring to the patient’s sex chromosome genotype (i.e., XX or XY).

Response: We carefully checked our manuscript text and supplementary files, and corrected the expression with “Sex” instead of “gender” throughout the whole text, Supplementary Figs. 1 and 3, and Supplementary Tables 1-4.

13. Repetitive text at end of page 2 in Glossary “Neuropathological evaluation” section.

Response: We rewrote the Glossary “Neuropathological evaluation” section, and deleted the repetitive text.

14. Please include additional references throughout glossary where appropriate when presenting previously published work. Specific examples include the genotype-enzymatic activity on page 1, Cognitive functional assessment on page 4, A β 40 and A β 42 generation on page 5,

Response: We checked and added relative references in the terms throughout the Glossary file.

15. Supplementary File Table i. please remove the word “change” in table and include accession number.

Response: We removed the word “change” and added “Accession number” in the Supplementary File Table i.

16. Extended Data Fig 1. Please include a detailed description of abbreviations where appropriate in the Figure legend.

Response: We added the detailed description of abbreviations in the Supplementary Figs. 1 and 3 legends.

17. Supplementary Table 4. What is NC?

Response: In Supplementary Table 4, the “NC” should be “CTRL”, represents the healthy control donors. We revised to “CTRL”.

18. A small formatting comment for supplementary table that goes across 4 pages (Supplementary table 1?), please put headings on each page and table legend on pdf.

Response: We added the headings and table legend of Supplementary Table 1 in each page.

Reviewer #3 (Remarks to the Author):

Review of “Aldehyde dehydrogenase 2 rs671 variant enhances Alzheimer’s disease pathology” by Wang et al.

The authors have in a screen of 329 human brain samples of investigated the role of ALDH2 rs671 polymorphism in AD. This genetic variant has previously been shown to present a risk for hypertension, diabetes and coronary heart disease in the Asian population. For AD, the impact of this polymorphism have so far been contradictory.

The investigation was initiated on a population biobank consisting of 59% male and 41% female samples, on which genotyping was conducted. In turn, amyloid plaque assessment was done to find potential correlation between alleles and phenotype. Further, detailed analysis on Ab levels was done with ELISA. Next, the authors investigated the effects of the Aldh2-gene on Ab-peptide ratios in the APPSwe mouse model of AD. This was done both with a knock-out and by pharmacological modulation of the Aldh2. The author then investigate the potential accumulation of 4-HNE on Ab-peptide ratios. Adduction of 4-HNE to Ab-peptides are found with mass spectrometry, and phenotypic effects were observed with an in vivo assay using HEK293 cells. Overall the manuscript present many interesting findings and connect a large screening study with detailed analysis of molecular mechanisms that highlight the importance of Aldh2 in AD.

However, I have major concerns regarding the proteomic data that the authors must address.

I am confused by how the authors present their proteomic data with volcano plots in extended data figs 6 and 14. It is not clear from the figures how many replicates were measured with proteomics. There are apparently no statistics involved in determining whether a protein has been up- or downregulated. Is extended data figure 6 based on only a single replicate of each condition? Rather, the authors choose an

arbitrary cutoff at $\text{abs}(\text{foldchange}) > 1.25$ in extended data figure 6 and $\text{abs}(\text{foldchange}) > 1.17$ in extended data figure 14. This is not an acceptable method to show significant effects in differential expression, because a protein with only $\text{logFC} = 0.5$ can have $\text{adj.p.value} > 0.0001$, but something $\text{logFC} = 3$ can have $\text{adj.p.value} = 0.5$. Instead, the authors must reprocess the data with proper statistical methods, such as Student's t-test or suitable moderated t-tests using either the R-packages limma or DESeq2. The appropriate statistical analysis must be done, including multiple testing correction such as the Benjamini-Hochberg procedure, where volcano plots should show logFC on the x-axis and $\text{log}_{10}(-\text{adj.p.value})$ on the y-axis.

Response: Thank you for your critical suggestion. In the Supplementary Fig. 6, Supplementary Fig. 14, and Fig. 3a, we have reorganized the presentation of proteomics data, including Venn diagrams, volcano plots, and bubble plots.

We conducted three replicates in each condition. And we used Venn diagrams to show the number of confident proteins identified in each of the three replicates for every condition.

As reported in the published papers, the limma (Linear Models for Microarray Data) package is commonly used for the analysis of gene expression data of microarray data, RNA-seq [1]. DESeq2 is a widely used package for differential analysis of RNA-seq data [2]. In this study, as suggested we used student's t-test to compare differential expression of proteins between two groups for the TMT-labelling quantitative proteomics data, followed by Benjamini-Hochberg multiple testing correction to calculate the adjusted p-value (adj.P).

Based on the recommendations and prior references reports, we set the threshold for differentially expressed proteins (DEPs) as: adjusted P value ≥ 0.05 , fold change ≤ 0.83 or ≥ 1.20 . In Supplementary Fig. 6 and Supplementary Fig.14, we re-presented the volcano plots with logFC on the x-axis and $-\text{log}_{10}(\text{adj.P value})$ on the y-axis to display the DEPs. Pathway analysis was performed on the DEPs, and the enriched signaling pathways of DEPs were visualized in the bubble plots.

All the revised figures are presented in the new Fig. 3a, Supplementary Fig.6 and Supplementary Fig. 14c, f-h (shown below) The revised description of results, discussion, and figure legends were highlighted in the revised manuscript files.

Supplementary Fig. 6 Proteomic analysis of N2a-APPswe cells pretreated with 60 μ M daidzin for 48 h.

Fig. 3a Bubble plot of KEGG pathways of differentially expressed proteins in N2a-APPswe cells pretreated with 60 μ M daidzin for 48 h.

c N2a-APPswe pretreated with Alda-1 (20 μ M)

Supplementary Fig. 14 (c) Proteomics analysis of N2a-APPswe pretreated with 20 μ M of Alda-1 for 24h. (f-h) Proteomics analysis of BV2 pretreated with 20 μ M of Alda-1 for 24h.

References:

- [1] Ritchie, M.E., et al. limma powers differential expression analyses for RNA-sequencing and microarray studies. *Nucleic Acids Res.* 20;43(7), e47 (2015). doi: 10.1093/nar/gkv007.
- [2] Liu, S., et al. Three Differential Expression Analysis Methods for RNA Sequencing: limma, EdgeR, DESeq2. *J. Vis. Exp.* 18, (2021). doi: 10.3791/62528.

Data availability: The proteomic data must be submitted to PRIDE (public repository). This has not been done.

Response: We deposited the proteomic data to the ProteomeXchange Consortium (<http://proteomecentral.proteomexchange.org>) via the iProX partner repository. And we got the dataset identifier PXD047209 and PXD047210. We added the statement in the “Data Availability” section of the manuscript.

Reviewer #1 (Remarks to the Author):

The content of all the corrections is OK. However, the text requires editing. For example,

P. 6, 'This conclusion remained consistent for both males and females (Supplementary Table 4).' 'This observation was consistent..' may be better

P 16. 'Overall, these findings indicated that ALDH2 deficiency disrupted the microglial functions to oA β , including attenuated production of inflammatory mediators, as well as impaired phagocytosis and clustering around A β plaque' should be 'Overall, these findings indicated that ALDH2 deficiency disrupted microglial functions in response to oA β , etc...'

P. 17 'Excessive ethanol exposure is detrimental to the brains and is a higher risk factor for AD.' To, for example: 'and increases the risk for AD.'

And 'Joshi et al. demonstrated aggravated neuropathology in ALDH2 AA-allele brains than GG brains after chronic alcohol consumption in mice.' to, for example 'Joshi et al. demonstrated worse neuropathology in the brains of ALDH2 AA-allele mice relative to GG (wild-type) mice after chronic alcohol consumption.'

P. 20. Change 'However, there are not enough AA individuals to establish AA as a risk factor for AD.' With, for example, 'However, the number of AA individuals is insufficient to determine whether AA is a risk factor for AD.'

P. 21 change 'These findings indicated that the positive correlation of ALDH2 rs671 polymorphism and A β pathology would be considered when applying antibodies for A β plaques or oligomers clearance ..' to 'These findings indicated that a positive correlation between ALDH2 rs671 polymorphism and A β pathology should be considered when applying antibodies for A β plaques ..'

P. 20, within the yellow paragraph, change twice 'wide-type' to 'wild-type'. Same correction also on page 79, twice.

Note also, that oral delivery of Alda-1 is not effective due to poor absorbance and first pass metabolism. For that reason, we usually delivered Alda-1 via subcutaneous Alzet pump.

Reviewer #2 (Remarks to the Author):

I appreciate the time the authors have taken to comprehensively address most of my major and minor comments.

However, major comment 4 from my initial review has not been adequately addressed in the main manuscript regarding assessment or identification of co-pathologies (either identified by the neuropathologist or routinely assessed by the neuropathologist but absent in each brain). The authors answered my comment by providing information about Ecog ratings, but not about other co-neuropathologies that may be present alongside the AB plaques reported here. Examples are given in Table 1 from <https://doi.org/10.1101/cshperspect.a028035> If no additional neuropathologies were assessed in the cohort, then this must be stated in the manuscript please, noting the caveat that other neuropathologies may be present in each brain, but were not assessed.

Additional comments:

1. Where has Table R1.1 been included in the manuscript? Please include as a new supplementary table.
2. Author Xue Wang has no affiliation.
3. When referring to any gene name throughout, please ensure the gene name is italicised.
4. On page 2, "14 AA alleles (3.0%), 123 GA alleles (26.2%), and 332 GG alleles (70.8%)" should be genotypes, not alleles.

5. Please remove "very obviously" from page 5.
6. Please consider an alternative to red font for highlighting statistically significant results in your supplementary tables. Red font next to black font is difficult for colourblind people to distinguish. Perhaps bold or italic may be more appropriate.
7. On page 6, please clarify that these transcriptome samples do not overlap samples in this study. Perhaps "However, transcriptome sequencing conducted on 50 additional population-matched postmortem human hippocampal tissues..."
8. Typo on page 17 - ethonal
9. Page 17/18 - Should Yukio, Kelly, Liu be Yukio *et al*, Kelly *et al* and Liu *et al*?

Reviewer #3 (Remarks to the Author):

With the revised manuscript the authors have addressed my previous concerns of their study. I believe this manuscript s now fit for publication.

Response letter

REVIEWER COMMENTS

Reviewer #1 (Remarks to the Author):

The content of all the corrections is OK. However, the text requires editing. For example, P. 6, ‘This conclusion remained consistent for both males and females (Supplementary Table 4).’ ‘This observation was consistent..’ may be better.

Response: Thanks very much for your kind suggestions for improving our manuscript language. We corrected and highlighted sentences in the revised manuscript as suggested. Here we revised this sentence to “This observation was consistent for both males and females (Supplementary Table 6)” and highlighted in the revised manuscript file.

P 16. ‘Overall, these findings indicated that ALDH2 deficiency disrupted the microglial functions to oA β , including attenuated production of inflammatory mediators, as well as impaired phagocytosis and clustering around A β plaque’ should be ‘Overall, these findings indicated that ALDH2 deficiency disrupted microglial functions in response to oA β , etc...’

Response: As suggested, this sentence in page 16 was revised to “Overall, these findings indicated that ALDH2 deficiency disrupted microglial functions in response to oA β , including attenuated production of inflammatory mediators, as well as impaired phagocytosis and clustering around A β plaques.”

P. 17 ‘Excessive ethanol exposure is detrimental to the brains and is a higher risk factor for AD.’ To, for example: ‘and increases the risk for AD.’

Response: This sentence in page 17 was revised to “Excessive ethanol exposure is detrimental to the brains and increases the risk for AD.”

And ‘Joshi et al. demonstrated aggravated neuropathology in ALDH2 AA-allele brains than GG brains after chronic alcohol consumption in mice.’ to, for example ‘Joshi et al. demonstrated worse neuropathology in the brains of ALDH2 AA-allele mice relative to GG (wild-type) mice after chronic alcohol consumption.’

Response: The sentence here was revised to “Joshi et al. demonstrated worse neuropathology in the brains of *ALDH2* AA-allele mice relative to GG (wild-type) mice after chronic alcohol consumption”.

P. 20. Change ‘However, there are not enough AA individuals to establish AA as a risk factor for AD.’ With, for example, ‘However, the number of AA individuals is insufficient to determine whether AA is a risk factor for AD.’

Response: The sentence here in page 20 was revised to “However, the number of AA individuals is insufficient to determine whether AA is a risk factor for AD”.

P. 21 change ‘These findings indicated that the positive correlation of *ALDH2* rs671 polymorphism and A β pathology would be considered when applying antibodies for A β plaques or oligomers clearance ..’ to ‘These findings indicated that a positive correlation between *ALDH2* rs671 polymorphism and A β pathology should be considered when applying antibodies for A β plaques ..’

Response: The revised sentence in page 21 was changed to “These findings indicated that a positive correlation between *ALDH2* rs671 polymorphism and A β pathology should be considered when applying antibodies for A β plaques or oligomers clearance ..”

P. 20, within the yellow paragraph, change twice ‘wide-type’ to ‘wild-type’. Same correction also on page 79, twice.

Response: We corrected the spelling of “wide-type” to “wild-type” in page 20 and page 79.

Note also, that oral delivery of Alda-1 is not effective due to poor absorbance and first pass metabolism. For that reason, we usually delivered Alda-1 via subcutaneous Alzet pump.

Response: Thanks very much for your explanation. We agree with your opinion and this explains the no significant alteration of A β plaque deposition in APP/PS1 mice induced by Alda-1 administrated *via* intragastric infusion. We added this explanation in the revised manuscript in page 20. The added sentence was as below:

Without ethanol consumption, we found Alda-1 administration also slightly reduced the plaque deposition area and mildly decreased the levels of A β 40 and A β 42 in APP/PS1 mice, with no significance, maybe due to poor oral delivery efficiency of Alda-1 (Supplementary Fig. 14a–b).

Reviewer #2 (Remarks to the Author):

I appreciate the time the authors have taken to comprehensively address most of my major and minor comments.

However, major comment 4 from my initial review has not been adequately addressed in the main manuscript regarding assessment or identification of co-pathologies (either identified by the neuropathologist or routinely assessed by the neuropathologist but absent in each brain). The authors answered my comment by providing information about Ecog ratings, but not about other co-neuropathologies that may be present alongside the AB plaques reported here. Examples are given in Table 1 from <https://doi.org/10.1101/cshperspect.a028035>. If no additional neuropathologies were assessed in the cohort, then this must be stated in the manuscript please, noting the caveat that other neuropathologies may be present in each brain, but were not assessed.

Response: Thanks for your comments and explanations. Besides ABC scores, other neuropathologies were also assessed for the donated postmortem brains, including Lewy bodies, Braak staging of Parkinson's disease, TDP-43 pathology, Primary age-related tauopathy, Cerebral amyloid angiopathy. We added these co-neuropathologies in the revised Supplementary Table 1, and conducted ordinal logistic regression analysis between these neuropathologies and *ALDH2* rs671 genotypes. The results were added in the revised Supplementary Table 4 and in the main text Result section. Detailed information of these neuropathologies were provided in the revised Supplementary File 1.

Additional comments:

1. Where has Table R1.1 been included in the manuscript? Please include as a new supplementary table.

Response: We added the original Table R1.1 into a new revised Supplementary Table 5, and renumbered the following supplementary tables in the revised manuscript and supplementary table file.

2. Author Xue Wang has no affiliation.

Response: Thanks for your correction. We added the indicated number of author Xue Wang's

affiliation.

3. When referring to any gene name throughout, please ensure the gene name is italicised.

Response: We carefully checked through the manuscript and supplementary files, and ensured all the gene names with italicized font.

4. On page 2, “14 AA alleles (3.0%), 123 GA alleles (26.2%), and 332 GG alleles (70.8%)” should be genotypes, not alleles.

Response: We corrected the expression and revised to “14 AA genotypes (3.0%), 123 GA genotypes (26.2%), and 332 GG genotypes (70.8%)”.

5. Please remove “very obviously” from page 5.

Response: We deleted “very obviously,” in page 5.

6. Please consider an alternative to red font for highlighting statistically significant results in your supplementary tables. Red font next to black font is difficult for colourblind people to distinguish. Perhaps bold or italic may be more appropriate.

Response: Thanks for your suggestion. In the revised Supplementary Tables 3 and 6-7, we used black bold font with ** for highlighting statistical significance.

7. On page 6, please clarify that these transcriptome samples do not overlap samples in this study. Perhaps “However, transcriptome sequencing conducted on 50 additional population-matched postmortem human hippocampal tissues...”

Response: Sorry for the confusion about the transcriptome samples. The 50 individuals for transcriptome were included in the cohort used for rs671 Sanger sequencing. Among the 50 individuals, 5 rs671-GG carriers overlapped those used for proteomic analysis. We added an additional column in Supplementary Table 1 to indicate the 50 individuals used for transcriptome and provided annotations in the main text.

8. Typo on page 17 – ethonal

Response: We corrected the spelling to “ethanol” in page 17.

9. Page 17/18 – Should Yukio, Kelly, Liu be Yukio et al, Kelly et al and Liu et al?

Response: We revised the indications of the reported references with authors in page 17-18. The revised expression was “Ando et al.” for reference 43, “Kelly et al.” for references 44-45, “Liu et al.” for reference 46.

Reviewer #3 (Remarks to the Author):

With the revised manuscript the authors have addressed my previous concerns of their study. I believe this manuscript's now fit for publication.

Response: We very appreciate for your great help for our manuscript.

Reviewer #1 (Remarks to the Author):

All the comments were addressed. No further revision is needed,

Reviewer #2 (Remarks to the Author):

This revision has addressed my previous comments.

A minor formatting edit: Please italicise "Primary age-related tauopathy" subheading on page 4 of Supplementary File 1.

Response to reviewers

REVIEWERS' COMMENTS

Reviewer #1 (Remarks to the Author):

All the comments were addressed. No further revision is needed.

Response: Thank you very much for your assistance in enhancing the quality of our manuscript.

Reviewer #2 (Remarks to the Author):

This revision has addressed my previous comments.

A minor formatting edit: Please italicise "Primary age-related tauopathy" subheading on page 4 of Supplementary File 1.

Response: We greatly appreciate for your correction. We italicised the “*Primary age-related tauopathy*” subheading in Supplementary Methods section on page 39 of the new version Supplementary Information file. Thank you again for your suggestions and assistance in enhancing the quality of our manuscript.